# PhysUniBench: A Multi-Modal Physics Reasoning Benchmark at Undergraduate Level

## Abstract

Physics problem-solving is a challenging domain for large AI models, requiring integration of conceptual understanding, mathematical reasoning, and interpretation of physical diagrams. Existing evaluations fail to capture the full breadth and complexity of undergraduate physics, whereas this level provides a rigorous yet standardized testbed for pedagogically relevant assessment of multi-step physical reasoning. To this end, we present **PhysUniBench**, a large-scale multimodal benchmark designed to evaluate and improve the reasoning capabilities of multimodal large language models (MLLMs) specifically on undergraduate-level physics problems. PhysUniBench consists of 3,304 physics questions spanning 8 major sub-disciplines of physics, each accompanied by one visual diagrams. The benchmark includes both open-ended and multiple-choice questions, systematically curated and difficulty-rated through an iterative model-in-the-loop process. The benchmark's construction involved a rigorous multi-stage process, including multiple roll-outs, expert-level evaluation, automated filtering of easily solved problems, and a nuanced difficulty grading system with five levels. Through extensive experiments, we observe that current state-of-the-art models encounter substantial challenges in physics reasoning. For example, GPT-5 achieves only about 53.7% accuracy in the proposed PhysUniBench. These results highlight that current MLLMs struggle with advanced physics reasoning, especially on multi-step problems and those requiring precise diagram interpretation. By providing a broad and rigorous assessment tool, PhysUniBench aims to drive progress in AI for Science, encouraging the development of models with stronger physical reasoning, problem-solving skills, and multimodal understanding. The benchmark and evaluation scripts are available at `https://anonymous.4open.science/r/PhysUniBenchmark-5784`.

## 1 Introduction

Physics, as a foundational science, is of paramount importance. The objectives of AI systems extend beyond mere information processing (Zhou et al., 2025) but encompass the attainment of complex reasoning, the resolution of challenging problems, and ultimately the facilitation of scientific discovery (Yang et al., 2024b; Liu et al., 2025). The ability to solve physics problems is a key indicator of such advanced reasoning capabilities. In recent years, state-of-the-art Large Language Models (LLMs) have achieved impressive results across a wide range of scientific domains (Xia et al., 2025b; Yang et al., 2024a; Li et al., 2024b; Tan et al., 2025; Xu et al., 2025a), such as attaining human-level accuracy on Olympiad-level mathematical problems (MAA, 2024; Rein et al., 2024; Hendrycks et al., 2021; He et al., 2024). Meanwhile, emerging Multimodal Large Language Models (MLLMs), such as GPT-4o (Hurst et al., 2024), Qwen2.5-VL (Bai et al., 2025b), InternVL-3 (Zhu et al., 2025), and Claude-3.7-Sonnet (Anthropic, 2025), integrate visual understanding and reasoning capabilities, allowing broader scientific applications (Ye et al., 2024; Hu et al., 2024; Xia et al., 2025a; Li et al., 2025; Wang et al., 2025; Zhao et al., 2025). However, their proficiency in physics domains remains an active area of research and evaluation.

Physics reasoning differs fundamentally from mathematical reasoning or factual question answering, as it requires the integration of domain knowledge, symbolic manipulation, real-world constraints, and the application of abstract physical principles to concrete and often visual scenarios. While MLLMs demonstrate strong performance in mathematics, they continue to struggle with physics

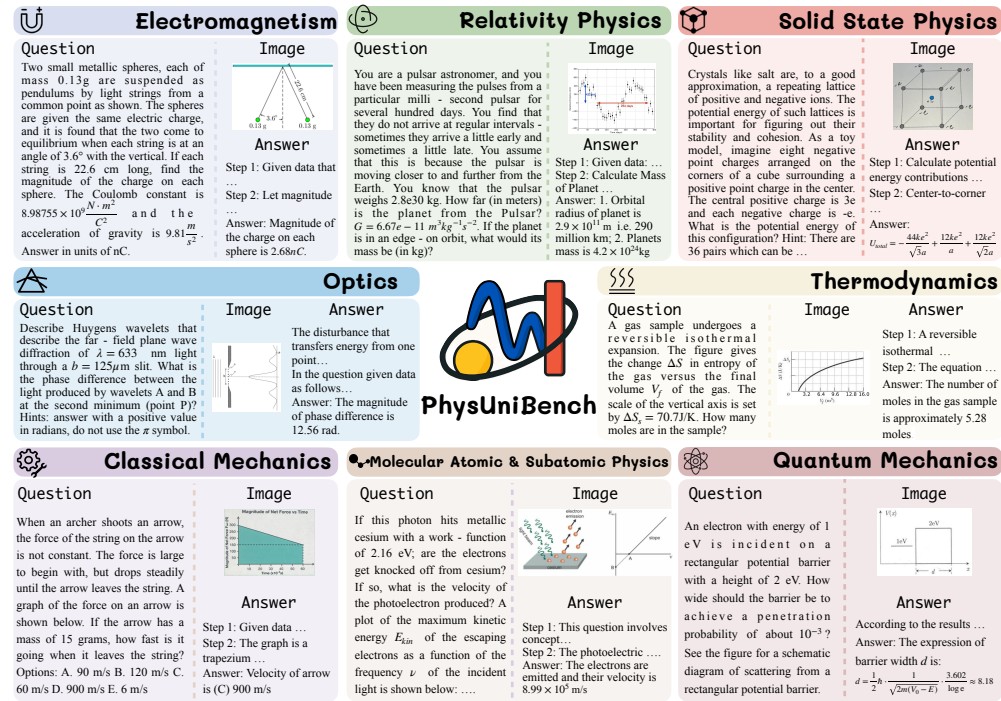

Figure 1: PhysUniBench is the first large-scale multimodal benchmark designed for evaluating undergraduate-level physics understanding, reasoning, and problem-solving. It includes 3,304 rigorously curated questions with accompanying diagrams, spanning eight core sub-disciplines from authentic university curricula.

reasoning. For example, GPT-4V achieves only 10.74% accuracy on physics questions in Olympiad-Bench (He et al., 2024), and the best model (OpenAI, 2024) in UGPhysics (Xu et al., 2025b) attains 49.8% accuracy. These results indicate that physics problems often require deeper and more integrated forms of reasoning, posing a fosrmidable challenge to current models. Therefore, a benchmark that rigorously evaluates these capabilities, particularly in visual and context-rich settings, is essential for advancing physics model development.

Current physics evaluations mainly focus on K12 (Zhang et al., 2025) or Olympiad-level problems (He et al., 2024; Huang et al., 2024). Although these benchmarks are valuable for assessing foundational knowledge and high-level problem-solving capabilities, they do not adequately represent the depth and diversity of reasoning cultivated in undergraduate physics education. The undergraduate curriculum plays a pivotal role in developing comprehensive conceptual frameworks and applied problem-solving abilities essential for training future scientists and engineers. Consequently, a benchmark at this level is not only crucial for assessing and advancing AI models' capacity for complex, curriculum-aligned multimodal physics reasoning, but also ensures alignment with established educational assessment practices in physics education (McDermott & Redish, 1999; Heller et al., 1992; Redish & Burciaga, 2003). UGPhysics (Xu et al., 2025b) represents a notable effort toward assessing undergraduate-level physics; however, its current iteration is limited to text-based problems and lacks visual components, overlooking the critical role of diagrammatic reasoning that is essential in real-world physics problem-solving and applications. In real-world physics problem-solving, diagrams play a central role in representing spatial relationships, experimental setups, and conceptual models. The ability to interpret and integrate visual information with textual reasoning is fundamental to mastering the discipline.

To address these issues, we introduce **PhysUniBench**, the first large-scale physics benchmark specifically designed for multimodal understanding, reasoning, and problem-solving at the undergraduate level. PhysUniBench comprises a total of *3,304 carefully curated physics problems*, each paired with an accompanying diagram to support the evaluation of *joint visual and textual reasoning capabilities*. All questions are sourced from authentic undergraduate physics curricula, ensuring both

academic rigor and content relevance. The benchmark spans *eight core sub-disciplines of physics*, including optics, electromagnetism, classical mechanics, quantum mechanics, relativity physics, solid state physics, thermodynamics and molecular, atomic & subatomic physics. To the best of our knowledge, it is the first undergraduate-level physics benchmark at this scale of diagrammatic richness, enabling a thorough assessment of multi-modal physics reasoning ability. To facilitate detailed analysis of model performance, each problem is annotated with a fine-grained difficulty level ranging from 1 to 5, following a rigorous multi-phase curation and calibration process. Both *open-ended (OE) and multiple-choice (MC) question formats* are included, allowing comprehensive assessment of diverse types of reasoning. Additionally, the inclusion of problems in *both Chinese and English* supports multilingual evaluation and enhances the benchmark's linguistic diversity. Benchmark examples are illustrated in Figure 1.

We conduct extensive evaluations of state-of-the-art MLLMs on PhysUniBench. The results reveal that these undergraduate-level multimodal physics problems remain highly challenging for current models. The best-performing model, GPT-5, achieves 53.7% overall accuracy on OE questions. However, performance drops sharply for most models, often falling below 10% on certain sub-disciplines and at higher difficulty levels. Significant performance disparities are observed across sub-domains and difficulty levels, highlighting existing limitations in MLLMs' ability to integrate physics knowledge, symbolic reasoning, and visual understanding. Given its scale, diversity, and rigor, PhysUniBench provides a valuable testbed for advancing future multimodal models with stronger scientific reasoning capabilities. Our main contributions are summarized as follows:

- We present PhysUniBench, the first large-scale undergraduate-level multimodal physics benchmark, consisting of 3,304 human-verified problems with accompanying diagrams.

- A systematically curated dataset spanning 8 core sub-disciplines, with multilingual support and fine-grained difficulty annotations, is provided to enable detailed physics reasoning evaluation.

- Extensive evaluation of state-of-the-art MLLMs are conducted, revealing significant challenges in multimodal physics reasoning and providing insights to guide future model development.

## 2 RELATED WORK

### 2.1 PHYSICS-SPECIFIC BENCHMARKS

Early physics benchmarks were typically embedded within broader scientific datasets. Particularly, MMLU-Pro (Wang et al., 2024e) targeted college-level knowledge, while ScienceQA (Lu et al., 2022) combined text and image inputs across diverse scientific subjects. GPQA (Rein et al., 2024) introduced graduate-level STEM questions designed to challenge retrieval-based methods. More recently, benchmarks focusing specifically on physics reasoning have emerged (Qiu et al., 2025; Xu et al., 2025b; Dai et al., 2025; Shen et al., 2025b; Zhang et al., 2025; Yu et al., 2025; Xiang et al., 2025). OlympiadBench (He et al., 2024) compiled thousands of bilingual Olympiad-level problems and HiPho (Yu et al., 2025) similarly collected 360 elite high-school Olympiad problems with human-aligned grading and medal-based comparison to top student performance. PhysicsArena (Dai et al., 2025) introduced a multimodal benchmark covering variable identification, process formulation, and solution derivation. PhyX (Shen et al., 2025b) and PhysReason (Zhang et al., 2025) contributed benchmarks focused on realistic scenarios and multi-step reasoning. PHYBench (Qiu et al., 2025) curated 500 original problems to mitigate data contamination, and UGPhysics (Xu et al., 2025b) assembled 5,520 undergraduate-level problems across thirteen subject areas. More recently, SeePhys (Xiang et al., 2025) developed a vision-focused benchmarks spanning from middle school to PhD to assess vision-essential physics diagram understanding.

Despite this progress, large-scale multimodal benchmarks for physics remain limited, particularly those with calibrated difficulty, broad sub-disciplinary coverage, and multilingual support aligned with university curricula. To address these gaps, we introduce PhysUniBench, a large-scale undergraduate-level multilingual physics benchmark that provides the most comprehensive testbed to date for evaluating scientific reasoning and multimodal understanding in state-of-the-art models.

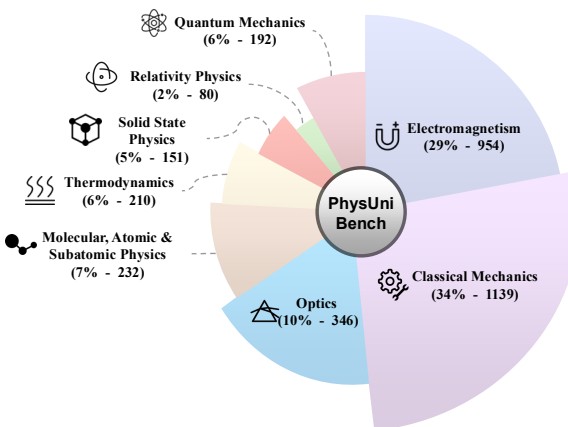

Figure 2: Distribution of PhysUniBench.

Table 1: Key Statistics of PhysUniBench.

| Statistic | Number |
|---|---|
| **Total questions** | **3304** |
|   - Multiple-choice questions | 1247 |
|   - Open-ended questions | 2057 |
| **Unique number of images** | **3304** |
| Difficulty level-1 questions | 663 |
| Difficulty level-2 questions | 661 |
| Difficulty level-3 questions | 660 |
| Difficulty level-4 questions | 661 |
| Difficulty level-5 questions | 659 |
| Average question tokens | 150.7 |
| Average option tokens | 184.0 |
| Average answer tokens | 441.9 |

## 2.2 MULTIMODAL LARGE LANGUAGE MODELS

Recent years have witnessed rapid advances in MLLMs that integrate visual and textual information. Early breakthroughs such as Flamingo (Alayrac et al., 2022) and PaLI (Chen et al., 2023) demonstrated that combining pretrained language models with visual encoders enables strong performance on diverse multimodal tasks. The release of GPT-4 (OpenAI & Achiam, 2024) further mainstreamed this capability, achieving near human-level results across many academic and professional benchmarks. Open-source efforts quickly followed, focusing on efficient architectural alignment between vision and language components. BLIP-2 (Li et al., 2023) employed a lightweight query transformer to bridge frozen vision and language models, while MiniGPT-4 (Zhu et al., 2024) showed that minimal adaptation suffices to elicit rich multimodal capabilities. More recently, LLaVA (Liu et al., 2024b;a; Li et al., 2024a), CogVLM (Wang et al., 2024d), Qwen-VL (Wang et al., 2024c; Bai et al., 2023; 2025a), InternVL (Chen et al., 2024; Zhu et al., 2025), and DeepSeek-VL (Lu et al., 2024a) further enhanced visual reasoning and language fluency through improved vision-language pretraining and hybrid architectures.

These advances reflect a clear trend toward instruction-tuned MLLMs capable of operating across modalities within a shared semantic space. However, challenges remain in fine-grained scientific reasoning, particularly in tasks requiring precise integration of visual, mathematical, and conceptual understanding (Fu et al., 2023; Ye et al., 2024; Lu et al., 2024b; Xia et al., 2024a;b; Yue et al., 2024). Addressing this gap, the PhysUniBench benchmark introduced in this work provides a comprehensive testbed for evaluating the reasoning capabilities of state-of-the-art MLLMs on complex, multimodal physics problems.

## 3 PHYSUNIBENCH

### 3.1 OVERVIEW OF PHYSUNIBENCH

**PhysUniBench** is a large-scale, multimodal benchmark specifically designed to evaluate the advanced reasoning capabilities of MLLMs on undergraduate-level physics problems. It aims to fill a critical gap in current benchmark ecosystems by offering a challenging, diverse, and diagnostic dataset that reflects the complexity and multimodal nature of real-world scientific problem solving.

Unlike prior benchmarks that focus on text-only math or physics tasks, PhysUniBench emphasizes *multimodal scientific reasoning*: all questions are paired with visual diagrams, requiring models to integrate textual and visual information to arrive at correct answers. This makes PhysUniBench uniquely suited to test the limits of current MLLMs in performing concept-rich, symbol-heavy, and context-dependent reasoning. The benchmark comprises a total of **3,304 problems**, divided into:

- **2057 open-ended questions** (OE format), requiring free-form answers that test the model's generation and justification capabilities.

Table 2: Comparison of Physics-Related Benchmarks. For general benchmarks, we report physics subset. **Image Num**: Count of problems with image. **Question Type:** OE: Open-ended, MC: Multiple-choice, FB: Fill-in-the-blank, J: Judgement. **Language Type:** EN: English, ZH: Chinese. **Knowledge Level:** K12: Elementary to High School; CEE: College Entrance Examination; COMP: Competition; COL: College; UG: Undergraduate; G: Graduate; Ph.D: Doctor of Philosophy.

| Benchmark | Size | Image Num | Multimodal | Difficulty Split | Know. Level | Question Type | Language Type |
|---|---|---|---|---|---|---|---|
| MMLU (Hendrycks et al., 2020) | 629 | 0 | ✗ | ✓ | K12/UG | MC | EN |
| AGIEval (Zhong et al., 2023) | 200 | 0 | ✗ | ✗ | CEE | MC, FB | EN |
| SciBench (Sun et al., 2024) | 64 | 64 | ✓ | ✗ | COL | OE | EN |
| SciEval (Sun et al., 2024) | 1657 | 0 | ✗ | ✓ | – | MC, J, FB | EN |
| GPQA (Rein et al., 2024) | 227 | 0 | ✗ | ✗ | PhD | MC | EN |
| MMMU (Yue et al., 2024) | 983 | 443 | ✓ | ✗ | COL | MC, OE | EN |
| OlympicArena (Huang et al., 2024) | 944 | 944 | ✓ | ✓ | COMP | MC, OE | EN |
| OlympiadBench (He et al., 2024) | 1958 | 1958 | ✓ | ✗ | COMP | OE | EN,ZH |
| EMMA (Hao et al., 2025) | 156 | 156 | ✓ | ✗ | CEE | MC, OE | EN |
| PHYBench (Qiu et al., 2025) | 500 | 0 | ✗ | ✓ | K12/UG/COMP | OE | EN |
| PhysReason (Zhang et al., 2025) | 1200 | 972 | ✓ | ✓ | K12/COMP | OE | EN |
| PHYSICSARENA (Dai et al., 2025) | 5103 | 5103 | ✓ | ✗ | CEE | OE | EN |
| PHYX (Shen et al., 2025b) | 3000 | 3000 | ✓ | ✗ | UG/G | MC, OE | EN |
| UGPhysics (Xu et al., 2025b) | 5520 | 0 | ✗ | ✓ | UG | MC,OE,J,FB | EN,ZH |
| PHYSICS (Feng et al., 2025) | 1297 | 289 | ✓ | ✗ | COL | OE,MC | EN |
| **PhysUniBench (Ours)** | 3304 | 3304 | ✓ | ✓ | UG | MC, OE | EN,ZH |

- **1247 multiple-choice questions** (MC format), constructed by converting challenging OE items into multiple-choice format with model-generated distractors.

PhysUniBench spans 8 major subfields of university physics, including: (1) Classical Mechanics; (2) Electromagnetism; (3) Optics; (4) Molecular, Atomic, and Subatomic Physics; (5) Thermodynamics; (6) Quantum Mechanics; (7) Solid State Physics; (8) Relativity Physics. The problems in PhysUniBench are meticulously curated from resources aligned with undergraduate physics curricula to facilitate a broad evaluation of a model's physics knowledge and reasoning skills. A detailed breakdown of the benchmark is provided in Figure 3 and Table 1. To ensure a discriminative evaluation, all problems in PhysUniBench are annotated with a difficulty level from 1 to 5, calibrated based on the performance of a strong baseline MLLM (e.g., Qwen2.5-VL-72B (Bai et al., 2025b)) through a 16-sample roll-out protocol. Problems the model solved trivially were removed to raise the difficulty floor; the rest are binned by model pass rate: the easiest top 0–20% are labeled 1, 20–40% labeled 2, 40–60% labeled 3, 60–80% labeled 4, and 80–100% labeled 5.

**Comparison with Existing Benchmarks.** Compared to existing benchmarks (see Table 2), PhysUniBench is distinguished by its focus on undergraduate-level physics, a large collection of multimodal questions across 8 core sub-disciplines, and fine-grained diversity in difficulty, question format, and language. While UGPhysics (Xu et al., 2025b) focuses on undergraduate-level content with abundant questions, it lacks multimodal elements essential for evaluating visual-textual reasoning. Meanwhile, PhysicsArena (Dai et al., 2025) offers extensive multimodal data but spans broad difficulty levels and educational stages, resulting in limited undergraduate-level coverage and reduced effectiveness for targeted evaluation. PhyX (Shen et al., 2025b) offers a diverse set of multimodal questions with a focus on undergraduate and graduate levels, but it lacks clearly defined difficulty stratification and multilingual support. In contrast, our PhysUniBench focuses explicitly on undergraduate physics, providing 3,304 human-verified multimodal problems, systematically stratified across five difficulty levels, covering eight major sub-disciplines, and supporting both English and Chinese. These features collectively make it a rigorous and versatile benchmark for advancing multimodal scientific reasoning in physics.

### 3.2 BENCHMARK CURATION PROCESS

**Data Acquisition.** PhysUniBench was constructed from a large-scale dataset of undergraduate-level physics problems from textbooks, exams, exercises, and competitions, selected to reflect typical undergraduate curricula. The curation prioritized problems requiring conceptual understanding, application of physical laws, and multi-step reasoning, while avoiding simple recall or plug-and-chug tasks. Overall clarity and unambiguous solutions were also key selection criteria. For PDF sources, we applied MinerU (Wang et al., 2024a) to parse problems into structured texts and images.

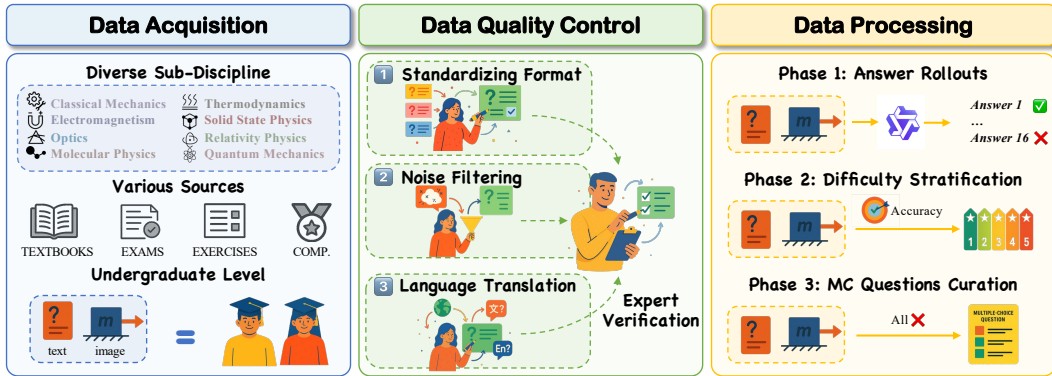

Figure 3: PhysUniBench is constructed through a rigorous three-stage data curation process designed to ensure high-quality multimodal physics problems. This pipeline systematically curates a wide range of questions across 8 core physics disciplines.

**Data Quality Control.** To ensure the clarity and consistency of PhysUniBench, we designed a quality control process during post-processing. First, all problems were reformulated to ensure they are phrased explicitly as questions and include sufficient contextual information for standalone interpretation. This step aimed to standardize question format and prevent ambiguity. Second, we removed redundant or irrelevant elements such as image numbering, cross-references, or formatting artifacts that do not contribute to problem understanding. These refinements help reduce noise and improve the readability and focus of each problem, thereby enhancing the overall quality and usability of the benchmark. Finally, to ensure language consistency, all problems not originally in English or Chinese were carefully translated into one of these two languages. Translations were manually verified to preserve the original meaning, technical accuracy, and problem structure, ensuring that the benchmark remains faithful to its source material while maintaining clarity for multilingual evaluation.

**Data Processing.** A distinctive feature of PhysUniBench is its multi-phase construction pipeline (Figure 3), which leverages advanced AI models for answer generation, evaluation, and difficulty calibration. This iterative approach systematically filters out trivial problems and produces a carefully stratified dataset that can more effectively probe the limits of current MLLMs in multimodal scientific reasoning.

The benchmark construction followed a three-phase process. In Phase 1, each source question was answered 16 times by Qwen2.5-VL-72B (Bai et al., 2025b), selected for its strong instruction-following and multimodal reasoning abilities. Answers were evaluated by GPT-4o (Hurst et al., 2024) for semantic or numerical correctness. Questions consistently answered correctly were removed to ensure a higher difficulty baseline, while the remaining answers formed a diverse pool for further analysis. In Phase 2, unsolved questions were stratified into five difficulty levels based on model accuracy and verified for correctness, forming the open-ended (OE) set designed to capture varying reasoning complexity. Phase 3 focused on the hardest questions which are never solved in Phase 1. They were reformulated into multiple-choice (MC) format using the model's own incorrect responses as distractors to reflect typical failure modes. These MC questions were also evaluated through 16 roll-outs and difficulty-ranked for nuanced evaluation.

Through this multi-phase process, the final benchmark consists of 2,057 open-ended questions and 1,247 multiple-choice questions, each annotated with a difficulty level from 1 to 5 and labeled by sub-discipline. The distribution is summarized in Table 1. A more detailed explanation of the construction pipeline is provided in Appendix A.

## 4 EXPERIMENT

### 4.1 EVALUATION SETUP

**Baselines.** We evaluate various types of methods, where the LLMs include: Grok 3 (xAI, 2025), DeepSeek V3 (DeepSeek-AI, 2025), and the MLLMs include: GPT-4o (Hurst et al., 2024), GPT-

Table 3: Main results on our PhysUniBench evaluated by accuracy (%). **Abbreviations:** OP = Optics; MAS = Molecular, Atomic, and Subatomic Physics; ME = Mechanics; SP = Solid State Physics; TH = Thermodynamics and Statistical Physics; EM = Electromagnetism and Electrodynamics; RE = Relativity; QM = Quantum Mechanics. The highest and second-highest accuracies are highlighted in red and blue, respectively.

| Models | Overall | OP | MAS | ME | SP | TH | EM | RE | QM |
|---|---|---|---|---|---|---|---|---|---|
| Multi-Choice Questions (MC) | | | | | | | | | |
| *Large Language Models* | | | | | | | | | |
| Grok 3 | 31.1 | 39.1 | 31.1 | 35.0 | 23.5 | 25.0 | 26.8 | 15.2 | 34.0 |
| DeepSeek V3 | 34.1 | 39.1 | 43.2 | 36.3 | 29.4 | 28.1 | 30.7 | 18.2 | 35.8 |
| *Multimodal Large Language Models* | | | | | | | | | |
| GPT-5 | 63.6 | 64.7 | 78.4 | 62.9 | 64.7 | 59.4 | 59.9 | 75.8 | 69.8 |
| GPT-4o | 33.7 | 42.9 | 39.2 | 40.1 | 33.3 | 32.8 | 35.4 | 41.2 | 36.2 |
| GPT-4o-mini | 27.3 | 33.8 | 27.0 | 28.6 | 15.7 | 28.1 | 25.8 | 21.2 | 24.5 |
| GPT-o3 | 43.6 | 66.2 | 54.1 | 43.0 | 35.3 | 31.3 | 41.1 | 27.3 | 30.2 |
| GPT-o4-mini | 36.7 | 57.9 | 55.3 | 42.1 | 31.2 | 24.7 | 41.2 | 30.9 | 35.9 |
| Claude-3.5-Sonnet | 36.5 | 43.2 | 50.9 | 41.3 | 32.1 | 44.0 | 35.8 | 45.5 | 44.9 |
| Gemini-2.5-Pro | 26.5 | 27.8 | 29.7 | 26.6 | 25.5 | 25.0 | 24.7 | 24.3 | 35.9 |
| Qwen2.5-VL-72B | 33.4 | 31.9 | 32.9 | 40.8 | 23.9 | 26.9 | 29.8 | 38.3 | 33.9 |
| InternVL-3-38B | 33.6 | 41.3 | 41.9 | 37.6 | 21.6 | 26.6 | 29.2 | 12.1 | 32.1 |
| Open-Ended Questions (OE) | | | | | | | | | |
| *Large Language Models* | | | | | | | | | |
| Grok 3 | 22.9 | 40.8 | 25.3 | 29.2 | 10.0 | 2.7 | 22.5 | 2.1 | 0.7 |
| DeepSeek V3 | 19.6 | 34.3 | 25.3 | 26.2 | 7.0 | 3.4 | 17.2 | 2.1 | 0.0 |
| *Multimodal Large Language Models* | | | | | | | | | |
| GPT-5 | 53.7 | 44.6 | 50.0 | 58.2 | 49.0 | 54.1 | 55.3 | 48.9 | 48.2 |
| GPT-4o | 20.9 | 30.5 | 24.1 | 27.8 | 5.0 | 3.4 | 20.2 | 2.1 | 6.2 |
| GPT-4o-mini | 15.3 | 20.0 | 13.9 | 21.0 | 6.1 | 2.7 | 16.5 | 2.1 | 0.0 |
| GPT-o3 | 24.8 | 43.2 | 30.4 | 30.8 | 5.0 | 5.5 | 25.4 | 2.1 | 0.0 |
| GPT-o4-mini | 26.5 | 51.2 | 31.0 | 38.2 | 10.0 | 6.2 | 28.4 | 2.1 | 0.0 |
| Claude-3.5-Sonnet | 19.0 | 37.6 | 28.5 | 26.2 | 8.0 | 4.8 | 17.9 | 2.1 | 0.0 |
| Gemini-2.5-Pro | 25.5 | 49.3 | 31.0 | 35.2 | 6.0 | 4.1 | 23.7 | 2.1 | 0.0 |
| Qwen2.5-VL-72B | 23.7 | 38.5 | 29.1 | 29.5 | 9.0 | 5.5 | 21.4 | 2.1 | 0.0 |
| InternVL-3-38B | 17.7 | 27.4 | 17.9 | 21.0 | 5.5 | 3.4 | 19.3 | 2.1 | 0.0 |

4o-mini (Hurst et al., 2024), GPT-o3 (OpenAI, 2025), GPT-o4-mini (OpenAI, 2025), Claude-3.5-Sonnet (Anthropic, 2024), Qwen2.5-VL-72B (Bai et al., 2025b), Gemini-2.5-pro-preview (Team, 2025), InternVL-3-38B (Zhu et al., 2025).

**Evaluation Protocols.** We adopt a standardized protocol to ensure consistent and comparable assessment on PhysUniBench. Models are evaluated in a zero-shot setting, receiving both textual descriptions and associated images as input. For MC questions, evaluation is based on exact matching with the correct answer. For OE questions, models must output their final answer using the LaTeX \boxed{} format. Answers are verified through symbolic computation with `SymPy` for mathematical equivalence and a LLM judge with GPT-4o for reasoning and semantic correctness.

**Evaluation Metrics.** Model performance is reported in terms of accuracy, including overall accuracy across the entire benchmark, as well as accuracy broken down by physics sub-discipline, difficulty level, and question type (open-ended versus multiple-choice).

A detailed description of the full evaluation process is provided in Appendix C.

### 4.2 MAIN RESULTS

**PhysUniBench exposes significant challenges in multimodal physics reasoning while highlighting progress in open-source models.** As shown in Table 3, accuracy remains modest across both

Table 4: Model performance across different difficulty levels and question types by accuracy(%). The highest and second-highest accuracies are highlighted in red and blue, respectively.

| Model | Level 1 | | Level 2 | | Level 3 | | Level 4 | | Level 5 | |
|---|---|---|---|---|---|---|---|---|---|---|
| | MC | OE | MC | OE | MC | OE | MC | OE | MC | OE |
| *Large Language Models* | | | | | | | | | | |
| Grok 3 | 47.2 | 31.2 | 33.2 | 26.8 | 29.7 | 24.8 | 22.8 | 18.7 | 22.6 | 12.9 |
| Deepseek V3 | 48.4 | 30.3 | 39.2 | 23.8 | 32.1 | 21.2 | 27.6 | 14.8 | 23.0 | 7.8 |
| *Multimodal Large Language Models* | | | | | | | | | | |
| GPT-5 | 71.6 | 70.9 | 68.0 | 58.6 | 63.9 | 51.8 | 55.6 | 46.7 | 58.9 | 40.4 |
| GPT-4o | 54.7 | 35.8 | 41.2 | 24.1 | 36.1 | 17.5 | 31.6 | 14.1 | 27.1 | 10.5 |
| GPT-4o-mini | 39.2 | 27.8 | 28.4 | 23.8 | 23.7 | 15.5 | 21.6 | 10.3 | 23.8 | 10.6 |
| GPT-o3 | 51.2 | 28.8 | 47.2 | 28.0 | 41.8 | 25.8 | 38.8 | 25.3 | 39.1 | 16.1 |
| GPT-o4-mini | 49.7 | 37.5 | 47.2 | 33.3 | 40.5 | 32.1 | 36.8 | 25.1 | 37.5 | 18.0 |
| Claude-3.5-Sonnet | 54.9 | 32.9 | 48.8 | 27.0 | 33.3 | 19.5 | 36.5 | 14.4 | 36.5 | 8.8 |
| Gemini-2.5-Pro | 23.2 | 36.8 | 21.6 | 29.9 | 23.7 | 27.7 | 29.6 | 23.1 | 34.3 | 14.4 |
| Qwen2.5-VL-72B | 60.5 | 37.0 | 45.1 | 31.4 | 31.6 | 21.7 | 16.8 | 15.3 | 12.3 | 8.8 |
| InternVL-3-38B | 49.2 | 33.7 | 34.8 | 21.6 | 32.1 | 17.0 | 27.2 | 10.7 | 24.2 | 8.0 |

MC and OE questions, underscoring the inherent difficulty of complex multimodal reasoning in physics. The newly tested tested GPT-5 achieves the best overall performance, reaching 63.6% on MC and 53.7% on OE while the second best model achieving only 43.6% and 26.5%, respectively. This highlights PhysUniBench as a crucial benchmark for identifying current limitations and guiding the development of more capable systems. Notably, while closed-source models such as GPT-o4-mini achieves better overall performance, open-source models such as Qwen2.5-VL-72B are progressively narrowing the gap in certain sub-disciplines, achieving comparable accuracy on some MC questions. This trend signals promising progress in open-source scientific reasoning, though challenges persist, particularly in open-ended questions and advanced physics domains.

**Performance across sub-disciplines exhibits significant disparities.** Models perform relatively well in areas like Optics and Molecule Physics, likely due to alignment with pretraining distributions. In contrast, sub-disciplines such as Relativity, Thermodynamics, and Quantum Mechanics show notably poor performance where accuracy in Relativity remains below 3% and most models fail open-ended questions in Quantum Mechanics. These results underscore the need for targeted model development to handle the advanced concepts and multi-step reasoning required in these domains.

## 5 DISCUSSION

**Explicit reasoning facilitates the multi-modal physics problem solving but gap exists.** Unlike general question answering tasks, physics reasoning requires systematic decomposition of complex problems, alignment of visual and textual information, and the structured application of physical principles. Our evaluation shows that the reasoning-based model GPT-5 achieves the highest performance, consistent with the benefits of explicit reasoning observed in other scientific domains (Bercovich et al., 2025; Fallahpour et al., 2025). However, the 24.8 % and 26.5 % accuracies of GPT-o3 and GPT-o4-mini on OE reflects only a modest improvement over models without explicit reasoning capabilities. This underscores the unique challenges of physics reasoning and highlights the need for further advances in model architectures and training strategies to support structured, physics-aware reasoning in complex multimodal tasks.

**Performance across difficulty levels exhibits a well-distributed accuracy.** As problem difficulty increases shown in Table 4, model accuracy consistently declines, with the most pronounced drop observed at higher levels. In particular, Levels 4 and 5 pose significant challenges, with most models achieving only around 10 percent accuracy on OE questions at Level 5. This sharp decline indicates the rationality of adapting Qwen2.5-VL for difficulty stratification. It highlights the benchmark's ability to stress-test current models and reveals their limitations in complex reasoning. The stratified difficulty levels also enable fine-grained evaluation on specific subsets, facilitating more targeted

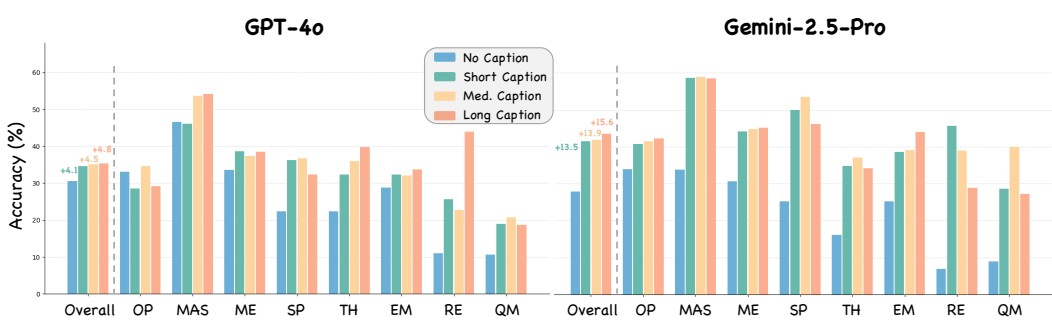

Figure 4: Performance changes for GPT-4o and Gemini-2.5-Pro when replacing image input with its corresponding caption labelled by GPT-4o beforehands.

analysis of model strengths and weaknesses across varying levels of problem complexity. A detailed analysis of difficulty levels is provided in Appendix B.

**Multimodal language model reasoning faces bottleneck on physics image understanding.** To analyze the potential causes on poor multi-modal physics reasoning, we replace the image input with GPT-4o generated captions, thereby isolating the contribution of visual information. We further vary the caption granularity (short: one sentence; medium: two–three sentences; long: highly detailed with examples shown in Appendix H) to test how textual richness affects performance. Surprisingly, performance improves consistently across nearly all sub-disciplines for both the strong model (GPT-4o) and the weaker (Gemini-2.5-Pro) as shown in Figure 4. This suggests a fundamental limitation in current MLLMs' ability to extract and reason over visual information in physics contexts. Closer inspection reveals that the benefit of captions is uneven across domains and models, suggesting differences in textual reasoning depth. For instance, detailed captions steadily enhance performance on **TH** for GPT-4o, yet yield unstable gains on **OP**. Conversely, Gemini-2.5-Pro derives disproportionately higher benefits in **MAS** than GPT-4o. Such divergence implies that while additional textual scaffolding can amplify reasoning, its utility is mediated by both the discipline-specific reasoning demands and the intrinsic strengths of each model. A possible explanation is that physics diagrams often convey abstract, symbolic, and spatial relations (e.g., forces, vectors, coordinate frames) that current visual encoders struggle to interpret accurately. In contrast, GPT-4o captions interpret this information explicitly in structured language, better suited to the models' strengths in textual reasoning. This observation generalizes beyond the possible reasoning clues embedded by GPT-4o within captions since single-sentence short captions also enhance better reasoning performance. The performance of LLM-only models such as Grok 3 and Deepseek V3 in Table 3 highlights the limitations of multimodal physics understanding, as these models achieve competitive results despite lacking access to visual information. These findings underscore the need for integrating physics-informed visual reasoning alongside textual reasoning to advance multimodal scientific understanding.

**Models mainly fail due to calculation errors despite having substantial physics knowledge.** To systematically characterize what are the typical error modes PhysUniBench, we conducted a qualitative error analysis on 212 incorrect answers produced by GPT-4o. With the assistance of graduate-level physics experts and a lightweight AI labeling tool, we classified errors into four categories (allowing for dual labeling): calculation errors (61.8%), reasoning logic errors (17.9%), problem-understanding errors (16.0%), and lack of prior physical knowledge (12.7%). This distribution identifies numerical and symbolic manipulation as the dominant failure mode. Conversely, the relatively low share of knowledge-related errors suggests that foundation models have internalized considerable physics knowledge through pretraining but lack effective mechanisms for context-specific retrieval and applicability. Based on these findings, we hypothesize program-aided (Wang et al., 2023) and tool-augmented approaches (Wang et al., 2024b) as potential directions for enhancing physics reasoning and developing next-gen physics enhanced MLLMs.

**Comparison between different LLM-as-judges.** Because LLM-as-Judge evaluation may introduce model-specific biases that could distort performance comparisons, it is essential to verify that our scoring remains stable under different evaluators. For open-ended evaluation, we adopt GPT-4o as our primary judge to remain consistent with prior benchmarks (Xu et al., 2025b). To assess potential judge-specific bias, we conducted a cross-judge sensitivity analysis comparing GPT-4o and

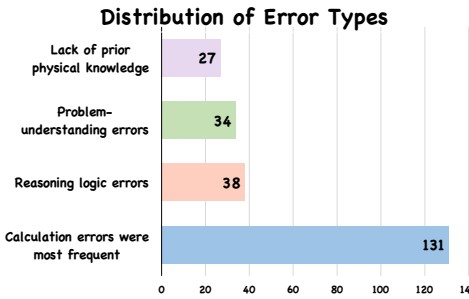

Figure 5: Four typical error types on open-ended questions of mechanics sub-discipline.

Table 5: Accuracy (%) of GPT-5 predictions on open-ended mechanics questions under two independent judges across difficulty levels.

| Difficulty Level | GPT-4o | Qwen2.5-VL-72B |
|---|---|---|
| D1 | 74.8 | 81.1 |
| D2 | 61.1 | 70.6 |
| D3 | 51.2 | 61.6 |
| D4 | 43.7 | 53.2 |
| D5 | 36.0 | 42.4 |
| **Overall** | **53.4** | **61.8** |

Qwen2.5-VL-72B when evaluating GPT-5 predictions in Table 5. Both judges exhibit the same monotonic accuracy decline from D1→D5 and preserve consistent model ranking across difficulty bins, indicating that our conclusions are not dependent on a single evaluator and that the scoring protocol is robust. In addition, human spot-checks confirm the alignment of LLM verdicts with physics-expert assessments, further supporting the validity of our evaluation approach.

**Limitations and future work.** Despite its comprehensive scope, PhysUniBench has several limitations. First, the use of Qwen2.5-VL-72B and GPT-4o in the data construction process may introduce subtle model-specific biases. Second, while automated evaluation of OE responses using advanced LLMs is scalable, it may not fully capture the nuance of expert human judgment. Third, the benchmark currently lacks coverage of certain physics topics, such as acoustics, which would be included to enhance topical completeness. Fourth, difficulty levels are defined across different knowledge areas; future work would incorporate varying difficulty levels within the same topic to better assess models' reasoning depth independent of content familiarity.

## 6 CONCLUSION

We present PhysUniBench, a large-scale undergraduate-level multimodal physics benchmark designed to rigorously evaluate the physics reasoning capabilities of MLLMs. It comprises 3,304 problems across 8 core sub-disciplines, integrating visual information and stratified difficulty through a multi-stage construction process. Our empirical results reveal that current state-of-the-art models struggle with deep physical reasoning, often relying on superficial cues rather than principled understanding. These findings underscore the need for next-generation models with stronger conceptual and multimodal physics reasoning abilities. PhysUniBench provides a comprehensive and challenging testbed to support future advancements in AI for science.

## 7 ETHICS STATEMENT

All authors have read and comply with the ICLR Code of Ethics. This work involves no human subjects or sensitive data, and we are unaware of any potential misuse, harm, or bias. No conflicts of interest or compromising sponsorships exist.

## 8 REPRODUCIBILITY STATEMENT

We have taken extensive steps to ensure the reproducibility of our work. The complete data curation pipeline is described in Section 3.2 and further detailed in the Appendix, providing transparency into the construction of our benchmark. All benchmark data and source code used to generate the results in this paper are released through an anonymous GitHub repository linked in the abstract, enabling full reproduction of experiments and analyses. Together, these resources allow researchers to verify our findings and extend them in future work.

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

# Appendix Table of Contents

## A    FURTHER DETAILS ON BENCHMARK CONSTRUCTION

### A.1    DATA SOURCING AND INITIAL COLLECTION

The problems in PhysUniBench were sourced from a large-scale dataset of undergraduate-level physics problems. This initial collection aimed to draw from materials representative of typical undergraduate physics curricula, covering a wide range of topics and problem styles encountered by students in their coursework and examinations. The selection criteria emphasized problems that require conceptual understanding, application of physical laws, and multi-step reasoning, rather than simple factual recall or plug-and-chug calculations. Clarity of problem statements and the existence of unambiguous solutions were also key considerations during the initial curation phase.

### A.2    BENCHMARK CONSTRUCTION

A distinctive feature of PhysUniBench is its multi-phase construction process, which leverages AI models for answer generation, evaluation, and difficulty calibration. This iterative approach was designed to filter out overly simplistic problems and to stratify the remaining ones by difficulty in a systematic manner.

**Phase 1: AI-Powered Answer Generation and Initial Filtering**

The first phase involved subjecting all initially collected questions to an extensive answer generation process. For each question, 16 independent answer roll-outs were conducted using the Qwen2.5-VL-72B model. This model was selected for its strong instruction-following capabilities and its proficiency in handling multimodal inputs, given that many problems included diagrams. The generation of 16 distinct roll-outs served multiple purposes: it allowed for an assessment of solution consistency, provided an opportunity to explore potentially diverse (yet correct) solution pathways, and generated a rich pool of answers for subsequent evaluation.

Following generation, each of the 16 answers for every problem was evaluated and matched by GPT-4o. The "matching" process involved determining if the generated answer was semantically equivalent to a known gold solution or numerically correct within a predefined tolerance. A critical filtering step was then applied: problems for which the Qwen2.5-VL-72B model correctly answered in all 16 roll-outs were removed from the benchmark. This decision was based on the premise that problems consistently solved by a capable multimodal LLM across multiple diverse attempts are likely to be relatively straightforward for the current generation of advanced models. By filtering out these "too easy" problems, PhysUniBench inherently establishes a higher difficulty floor, ensuring that the benchmark is not saturated by trivial questions and is better positioned to test the boundaries of model capabilities. This focuses the benchmark on material that presents a more substantial challenge, making it more effective for differentiating among high-performing models and for tracking meaningful progress in advanced AI reasoning.

**Phase 2: Difficulty Stratification for Open-Ended Questions**

For the problems that remained after the initial filtering, specifically, those that were not answered correctly in all 16 roll-outs by the Qwen2.5-VL-72B model, a difficulty level was subsequently assigned. This assignment was determined based on the accuracy achieved by Qwen2.5-VL-72B across its 16 roll-outs for each individual problem. Problems were then categorized into five difficulty levels, with Level 1 representing the easiest among the filtered set and Level 5 representing the most difficult. The mapping from roll-out accuracy to difficulty level was designed to produce an approximately balanced distribution of problems across the five levels, thereby providing a fine-grained scale for evaluating model performance. These curated problems form the open-ended question set within PhysUniBench.

**Phase 3: Conversion to Multiple-Choice Format for Consistently Incorrect Problems**

A special procedure was implemented for problems that Qwen2.5-VL-72B answered incorrectly in all 16 of its roll-outs. These problems, representing the most challenging set for the generation model, were converted into a MC format. The construction of these MC incorporated an innovative approach to distractor generation: three incorrect options (distractors) were randomly selected from the 16 incorrect answers produced by Qwen2.5-VL-72B for that specific problem. The correct answer (gold solution) was then added to form a four-option single-choice question. This method of

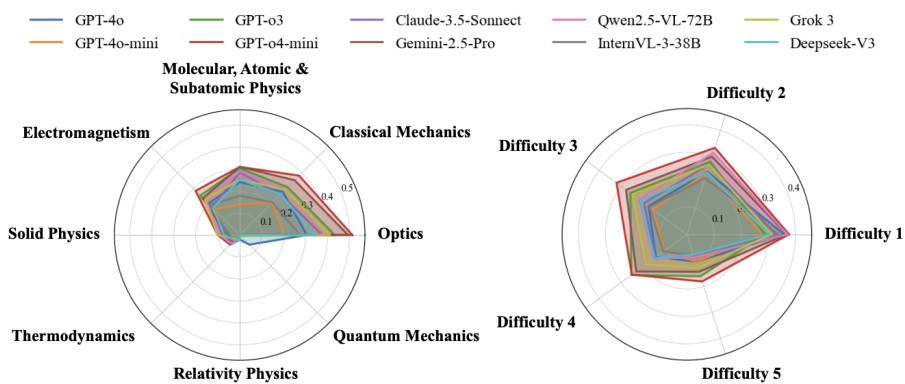

Figure 6: SOTA MLLMs performance on PhysUniBench (open-ended subset) by sub-discipline and difficulty, highlighting the significant challenges in multimodal physics reasoning.

leveraging a model's own failure modes to create distractors is intended to produce incorrect options that are plausible and diagnostically useful, as they reflect common model misconceptions or error patterns rather than arbitrary or easily identifiable incorrect choices. This enhances the quality of the MC portion of the benchmark, making it a more effective tool for diagnosing specific model weaknesses by testing whether a model can distinguish the correct answer from its own typical mistakes.

These newly formulated MC problems were then subjected to another 16 rounds of roll-outs using the same Qwen2.5-VL-72B model. Subsequently, they were filtered (if any proved too easy even in MC format, though this was expected to be rare given their origin) and graded by difficulty on the same 1-to-5 scale, based on the model's performance on the MC task.

This process resulted in a benchmark with 2,057 open-ended questions and 1,247 multiple-choice questions, all graded for difficulty. The distribution is shown in Table 1.

## B ADDITIONAL ANALYSIS

### B.1 RADAR PERFORMANCE COMPARISON

To further illustrate the multimodal reasoning capabilities of state-of-the-art MLLMs, we visualize model performance on the open-ended subset of PhysUniBench across two key axes: physics sub-disciplines (left radar chart) and question difficulty levels (right radar chart), as shown in Figure 6.

**Sub-discipline-level Comparison.** The left radar chart highlights model accuracy across eight major subfields of undergraduate physics. A notable trend is that models consistently perform better on Optics and Classical Mechanics, where problem types often involve more familiar visual elements and relatively intuitive reasoning steps. For instance, GPT-4o, GPT-4o-mini, and Gemini-2.5-Pro demonstrate relatively high accuracy on Optics, suggesting a degree of robustness in handling diagram-based ray-tracing or lens problems. In contrast, performance sharply drops on domains such as Thermodynamics, Solid State Physics, and Relativity Physics, with most models achieving near-random accuracy, highlighting their limited grasp of abstract physical processes or complex mathematical formulations.

**Difficulty-level Comparison.** The right radar chart reveals how model performance degrades as question difficulty increases. At Difficulty Level 1 and Level 2, several models, particularly GPT-5 and Qwen2.5-VL-72B, achieve moderate success, reflecting their ability to solve simpler conceptual or numerical problems. However, starting from Level 3, a steep decline in accuracy is observed across nearly all models. The lowest accuracy appears at Difficulty Level 5, where tasks often require multi-step reasoning, synthesis of visual and symbolic information, and deeper physical understanding.

**Overall Insights.** These radar plots collectively underscore two key challenges: (1) performance disparities across different physics domains due to varying levels of abstractness and visual complexity, and (2) difficulty sensitivity, where models falter significantly on higher-order reasoning tasks. The results affirm that while current MLLMs exhibit partial competence in lower-difficulty, visually grounded physics tasks, they remain far from achieving expert-level reasoning across the full spectrum of undergraduate physics problems.

## B.2 Impact of Image vs. Caption Inputs

**Image inputs show inconsistent benefits compared to caption inputs.** As summarized in Table 6, both GPT-4o and Gemini-2.5-Pro generally perform better when provided with caption descriptions rather than raw images. In particular, for open-ended (OE) questions, replacing images with captions leads to clear accuracy gains across most physics domains, suggesting that current multimodal models are not yet adept at extracting and reasoning over complex visual information. For example, GPT-4o's overall OE accuracy improves from 35.0% (image) to 37.4% (caption) in English, and Gemini-2.5-Pro shows even more pronounced gains in the Chinese OE setting (20.6% with image vs. 57.6% with caption). However, the effect is not uniform: in certain domains such as Optics and Solid State Physics, image inputs occasionally lead to small improvements in specific MC settings. These mixed results indicate that while visual data can contain valuable cues, current architectures often fail to reliably integrate them, highlighting a key avenue for future research in multimodal reasoning systems.

## B.3 Performance Across Difficulty Levels

**Higher difficulty levels significantly reduce model accuracy.** Table 7 shows that both GPT-4o and Gemini-2.5-Pro achieve strong performance on the easiest problems (Level 1) but exhibit a sharp accuracy decline as difficulty increases. For instance, GPT-4o's English OE accuracy on caption data drops from 67.7% at Level 1 to just 24.0% at Level 5, and a similar downward trend is observed across Chinese OE and MC tasks. This pattern persists when using image data, with particularly steep declines in Levels 4 and 5, suggesting that higher-order reasoning and abstraction remain significant challenges for current models. The consistent drop across modalities and languages underscores the benchmark's ability to expose reasoning limits beyond surface-level pattern recognition, providing valuable insight for guiding model improvements aimed at deeper scientific reasoning.

## B.4 Cross-Model Performance Analysis on Bilingual Task Sets

Table 8 presents the performance comparison of various large language models across Chinese and English physics tasks. Overall, GPT-5 achieves the best performance in both language settings, attaining an overall accuracy of 44.0% on English tasks and a remarkable 72.6% on Chinese tasks, significantly outperforming all other models. Notably, all models demonstrate superior performance on Chinese tasks compared to their English counterparts, which may be attributed to differences in dataset difficulty distribution or language-specific characteristics of the training corpora.

Regarding subject-specific performance, Optics (OP) and Modern Astrophysics (MAS) exhibit the most pronounced inter-model variations. GPT-5 achieves 76.9% accuracy on English optics tasks, substantially exceeding other models. However, most models struggle with advanced physics topics such as Relativity (RE) and Quantum Mechanics (QM) in English tasks, with accuracy rates approaching or equal to 0%, indicating that these domains impose higher demands on models' reasoning capabilities. In contrast, performance on Chinese tasks shows improvement in these advanced topics, though considerable room for enhancement remains. Claude-3.5-Sonnet and GPT-o3 demonstrate consistent performance on Chinese tasks with overall accuracies of 47.9% and 47.2% respectively, showcasing strong cross-lingual understanding capabilities.

## C Evaluation Protocols

To ensure consistent and comparable evaluations of model performance on PhysUniBench, we propose the following standardized protocol. This will enable an objective assessment of models'

Table 6: Combined results on caption and image data evaluated by accuracy (%). Abbreviations: OP = Optics; MAS = Molecular, Atomic and Subatomic Physics; ME = Mechanics; SP = Solid State Physics; TH = Thermodynamics and Statistical Physics; EM = Electromagnetism and Electrodynamics; RE = Relativity; QM = Quantum Mechanics.

| | GPT-4o | | | | Gemini-2.5-Pro | | | |
|---|---|---|---|---|---|---|---|---|
| Domain | EN-MC | CN-MC | EN-OE | CN-OE | EN-MC | CN-MC | EN-OE | CN-OE |
| | *Long Caption Data* | | | | | | | |
| Overall | 30.3 | 35.7 | 37.4 | 37.9 | 27.0 | 49.3 | 40.2 | 57.6 |
| OP | 15.4 | 34.2 | 37.9 | 29.4 | 46.2 | 40.0 | 31.0 | 51.6 |
| MAS | 100.0 | 34.7 | 50.0 | 31.8 | 100.0 | 56.9 | 25.0 | 52.0 |
| ME | 35.1 | 39.0 | 36.1 | 43.6 | 24.4 | 50.2 | 40.3 | 65.9 |
| SP | 36.4 | 20.0 | 25.0 | 47.7 | 36.4 | 40.0 | 50.0 | 58.0 |
| TH | 33.3 | 34.7 | 50.0 | 41.0 | 13.3 | 46.9 | 16.7 | 59.7 |
| EM | 25.8 | 35.6 | 38.8 | 34.8 | 28.0 | 50.0 | 42.2 | 55.9 |
| RE | 100.0 | 35.5 | 0.0 | 40.4 | 0.0 | 58.1 | 0.0 | 57.5 |
| QM | 0.0 | 35.9 | 0.0 | 38.9 | 0.0 | 56.6 | 0.0 | 52.5 |
| | *Image Data* | | | | | | | |
| Overall | 32.6 | 40.5 | 35.0 | 14.0 | 23.7 | 27.8 | 39.6 | 20.6 |
| OP | 23.1 | 45.0 | 34.5 | 29.9 | 30.8 | 27.5 | 24.1 | 53.3 |
| MAS | 100.0 | 37.5 | 25.0 | 24.0 | 50.0 | 29.2 | 25.0 | 31.2 |
| ME | 36.3 | 42.3 | 34.6 | 21.2 | 23.2 | 28.6 | 42.1 | 28.7 |
| SP | 27.3 | 35.0 | 25.0 | 2.3 | 45.5 | 20.0 | 33.3 | 2.3 |
| TH | 40.0 | 30.6 | 16.7 | 2.2 | 6.7 | 30.6 | 25.0 | 2.2 |
| EM | 29.1 | 41.1 | 37.1 | 8.1 | 23.6 | 25.7 | 39.2 | 12.6 |
| RE | 0.0 | 41.9 | 0.0 | 2.1 | 0.0 | 25.8 | 0.0 | 2.1 |
| QM | 0.0 | 36.2 | 0.0 | 6.2 | 0.0 | 35.9 | 0.0 | 0.0 |

Table 7: Combined results by difficulty levels evaluated by accuracy (%). Columns: EN-MC, CN-MC, EN-OE, CN-OE for GPT-4o and Gemini-2.5-Pro.

| | GPT-4o | | | | Gemini-2.5-Pro | | | |
|---|---|---|---|---|---|---|---|---|
| Difficulty | EN-MC | CN-MC | EN-OE | CN-OE | EN-MC | CN-MC | EN-OE | CN-OE |
| | *Long Caption Data* | | | | | | | |
| 1 | 57.0 | 52.6 | 67.7 | 67.5 | 24.1 | 57.9 | 67.7 | 78.0 |
| 2 | 30.4 | 40.9 | 45.2 | 46.0 | 24.1 | 52.1 | 46.0 | 63.2 |
| 3 | 30.8 | 28.7 | 24.8 | 37.4 | 26.9 | 46.8 | 36.8 | 50.7 |
| 4 | 15.2 | 31.0 | 24.6 | 20.7 | 27.9 | 42.7 | 27.8 | 47.4 |
| 5 | 18.0 | 25.3 | 24.0 | 17.8 | 32.1 | 47.1 | 22.4 | 48.6 |
| | *Image Data* | | | | | | | |
| 1 | 49.4 | 57.3 | 67.7 | 21.7 | 16.5 | 26.3 | 64.6 | 24.5 |
| 2 | 35.4 | 43.3 | 38.9 | 17.5 | 22.8 | 21.1 | 45.2 | 23.2 |
| 3 | 39.7 | 34.5 | 24.0 | 14.7 | 20.5 | 25.2 | 35.2 | 24.5 |
| 4 | 19.0 | 36.8 | 25.4 | 9.1 | 24.1 | 32.2 | 30.2 | 20.0 |
| 5 | 19.2 | 30.6 | 18.4 | 7.0 | 34.6 | 34.1 | 22.4 | 10.8 |

reasoning and problem-solving capabilities, ensuring fairness and reproducibility across different evaluations.

## C.1 EVALUATION SETTING

Models are evaluated in a zero-shot setting by default, where no prior examples are provided to the models and they must solve problems without contextual cues or demonstrations. For MLLMs that support few-shot prompting, performance under few-shot settings may also be reported, with a clear specification of the number of examples used in the report. This allows flexible benchmarking across different prompting strategies.

Table 8: Performance comparison across different models on Chinese and English tasks by subject (%).

| Models | Overall | OP | MAS | ME | SP | TH | EM | RE | QM |
|---|---|---|---|---|---|---|---|---|---|
| **English** | | | | | | | | | |
| GPT-4o | 32.6 | 23.1 | 100.0 | 36.3 | 27.3 | 40.0 | 29.1 | 0.0 | 0.0 |
| Claude-3.5-Sonnet | 26.2 | 30.8 | 100.0 | 25.6 | 36.4 | 46.7 | 23.6 | 0.0 | 0.0 |
| Qwen2.5-VL-72B | 28.0 | 30.8 | 100.0 | 29.8 | 36.4 | 20.0 | 25.8 | 0.0 | 0.0 |
| GPT-4o-mini | 30.5 | 23.1 | 100.0 | 32.1 | 27.3 | 40.0 | 28.6 | 0.0 | 0.0 |
| Gemini-2.5-Pro | 23.7 | 30.8 | 50.0 | 23.2 | 45.5 | 6.7 | 23.6 | 0.0 | 0.0 |
| GPT-o4-mini | 38.4 | 53.8 | 100.0 | 36.3 | 54.5 | 13.3 | 39.0 | 100.0 | 0.0 |
| InternVL-3-38B | 26.7 | 46.2 | 100.0 | 27.4 | 27.3 | 33.3 | 23.1 | 50.0 | 0.0 |
| GPT-o3 | 35.9 | 53.8 | 50.0 | 31.6 | 36.4 | 33.3 | 38.5 | 50.0 | 0.0 |
| Grok 3 | 27.0 | 30.8 | 100.0 | 29.2 | 36.4 | 26.7 | 23.6 | 0.0 | 0.0 |
| DeepSeek V3 | 25.2 | 30.8 | 100.0 | 27.4 | 27.3 | 26.7 | 22.0 | 0.0 | 0.0 |
| GPT-5 | **44.0** | **76.9** | **100.0** | **42.9** | 45.5 | 20.0 | **43.4** | **100.0** | 0.0 |
| **Chinese** | | | | | | | | | |
| GPT-4o | 40.5 | 45.0 | 37.5 | 42.3 | 35.0 | 30.6 | 41.1 | 41.9 | 36.2 |
| Claude-3.5-Sonnet | 47.9 | 45.0 | 51.4 | 52.8 | 32.5 | 42.9 | 47.5 | 48.4 | 44.9 |
| Qwen2.5-VL-72B | 35.3 | 31.7 | 31.9 | 41.9 | 20.0 | 28.6 | 35.6 | 38.7 | 33.9 |
| GPT-4o-mini | 25.9 | 35.0 | 25.0 | 26.5 | 12.5 | 24.5 | 23.3 | 22.6 | 24.5 |
| Gemini-2.5-Pro | 27.8 | 27.5 | 29.2 | 28.6 | 20.0 | 30.6 | 25.7 | 25.8 | 35.8 |
| GPT-o4-mini | 44.2 | 58.3 | 54.2 | 45.0 | 25.0 | 28.6 | 43.1 | 29.0 | 35.8 |
| InternVL-3-38B | 36.7 | 40.8 | 40.3 | 43.6 | 20.0 | 24.5 | 34.7 | 9.7 | 32.1 |
| GPT-o3 | 47.2 | 67.5 | 54.2 | 49.7 | 35.0 | 30.6 | 43.6 | 25.8 | 30.2 |
| Grok 3 | 33.0 | 40.0 | 29.2 | 38.3 | 20.0 | 24.5 | 29.7 | 16.1 | 34.0 |
| DeepSeek V3 | 38.2 | 40.0 | 41.7 | 41.5 | 30.0 | 28.6 | 38.6 | 19.4 | 35.8 |
| GPT-5 | **72.6** | 63.3 | **77.8** | **74.6** | **70.0** | **71.4** | **74.8** | **74.2** | **69.8** |

## C.2 INPUT FORMAT

MLLMs are provided with both the problem text and associated images as input. These models are expected to integrate visual and textual information to solve the problem. For text-only models, only the textual portion of each problem is provided.

## C.3 OUTPUT EVALUATION

Evaluation criteria differ based on question type. For multiple-choice questions, evaluation is straightforward: accuracy is determined by exact matching between the model's selected option and the correct answer. For open-ended questions, models are required to produce a final answer enclosed in LaTeX's \boxed{} format Feng et al. (2025). The correctness of the generated answers is assessed through a combination of symbolic computation and language model-based reasoning. Specifically, an exact match with the ground truth is first attempted. If the answer is expressed as a mathematical formula, symbolic computation tools such as SymPy are used to check for mathematical equivalence with the reference solution. If symbolic equivalence cannot be determined, or if the answer contains natural language components, an advanced language model, such as GPT-4, is used as a judge to assess the conceptual accuracy of the response. When ambiguous cases arise, or when critical problems require more nuanced assessment, human evaluation may be employed to supplement automated judgments.

## C.4 MC QUESTION EVALUATION LOGIC

In MC questions, the evaluation of open-ended answers similarly combines symbolic computation with language understanding. The model's final answer must appear in \boxed{} format. The evaluation process begins by attempting an exact match between the model's output and the reference answer. If the answer involves mathematical expressions, SymPy is employed to verify mathemat-

ical equivalence. When symbolic equivalence cannot be confirmed, conceptual accuracy is judged by a large language model, ensuring that the semantic meaning of the answer aligns with the correct solution. In cases where the model's output is ambiguous or clearly erroneous, human reviewers provide final validation.

## C.5 METRICS

The primary evaluation metric for PhysUniBench is accuracy. Results are reported across multiple dimensions to provide a comprehensive understanding of model performance. Specifically, we report overall accuracy across the entire benchmark, accuracy by physics sub-discipline as defined in Table 1, accuracy across five difficulty levels from Level 1 to Level 5, and accuracy by question type, distinguishing between open-ended and multiple-choice questions. This multi-dimensional reporting enables fine-grained analysis of model strengths and weaknesses.

## C.6 EVALUATION STEPS

The following steps outline the precise evaluation protocol:

- Step 1. Prediction Generation: Initially, the models generate predictions based on the provided input query, which incorporates problem descriptions and relevant images.

- Step 2. Answer Extraction: Raw model predictions may include reasoning steps, intermediate explanations, or irrelevant filler. To extract the definitive answer, we employ rule-based answer extraction strategies tailored to the type of problem. For open-ended questions, the goal is to extract the final numeric value, formula, or derived result while filtering out irrelevant text.

- Step 3. Automated Evaluation with LLM Judge: For OE questions, after extracting the answer, we compare it against the ground truth to determine its correctness. Since OE questions can have multiple valid answer forms, we use a language model evaluator, such as GPT-4, as a judge to assess conceptual accuracy. The evaluator is provided with the extracted answer and the ground truth solution. The evaluator's task is to determine if the extracted answer aligns with the expected solution, checking for both correctness and completeness. Multiple runs of the evaluator ensure robustness: the evaluator's decision-making process is tested across multiple attempts to ensure consistent results.

- Step 4. Evaluation for MC: For multiple-choice questions, we first attempt a direct match between the model's selected option and the correct answer. If the direct matching fails, the LLM evaluator as in OE questions will be employed to compare the model's reasoning and answer choice against the ground truth. This is done to confirm that the model's reasoning, even when misaligned with the correct answer, aligns logically with the correct options.

## C.7 HANDLING AMBIGUOUS OR CRITICAL SAMPLES

For particularly difficult, ambiguous, or critical samples, human evaluation is employed to provide an additional layer of judgment. This is necessary when automated evaluation yields uncertain results, when multiple valid interpretations exist, or when critical problems must be reviewed to ensure that model performance is assessed accurately and fairly.

## C.8 RESULTS REPORTING

All evaluation results are reported in terms of overall accuracy and broken down by difficulty level, question type, and sub-discipline. The reporting also includes qualitative insights into model strengths and weaknesses, highlighting areas where models struggle—such as particular sub-disciplines of physics or specific problem types—and providing analysis of common failure modes, including conceptual errors, diagram misinterpretations, and calculation mistakes. Additionally, we provide simulated baseline results, generated through random answer selection, to serve as a point of comparison for model performance on the benchmark.

## D  PROMPTS USED IN PHYSUNIBENCH

This section presents prompts used in our benchmark including answering prompt for multiple-choice question (MC) and open-ended (OE) formats as well as prompts for captioning physics diagram in discussion. Each format is designed with consistent instructions to ensure clarity in model evaluation.

---

**Answering Prompt: Multiple-choice (MC)**

**Instruction Prompt (MC).** You are a helpful assistant. Based on the following question and options, choose the most appropriate answer. The image is provided separately.

**Question:** {question}

**Options:** {options_prompt}

**Expected Response:** Please respond with only the letter of the correct answer (A, B, C, or D).

---

**Answering Prompt: Open-ended (OE)**

**Instruction Prompt (Open-ended).** You are a physics expert assistant. Solve the following question step-by-step. At the VERY END of your answer, output ONLY the FINAL ANSWER in this format:

$$\boxed{\text{your\_final\_answer\_here}}$$

You MUST put the final answer in the \boxed{} environment. Do NOT include multiple boxes. Do NOT include \boxed{} anywhere else in your reasoning. The box must appear on the last line of the response.

**Question:** {question}
**Answer:** (model should generate full reasoning followed by one boxed final answer)

---

**Long Captioning Prompt (EN)**

You are a physics teaching assistant. Below is an image related to a physics problem. Write a detailed caption (multiple sentences OK) that:

1. Clearly names and labels all key elements or variables shown (e.g. masses, forces, fields, angles).

2. Thoroughly describes the physical scenario or phenomenon (e.g. "a block of mass m sliding down a frictional incline of angle $\theta$," "electric field lines emanating from a point charge").

3. Explains which aspects are critical for solving a related physics question (e.g. "resolve weight into components parallel and perpendicular to the surface," "note the separation distance r for Coulomb's law").

4. Provides any context or background needed to understand why the setup matters.

---

**Long Captioning Prompt (CN)**

您是一名物理教学助理。下面是一幅与物理问题相关的图像。请为其撰写一段详细的说明文字（可使用多句），要求：

1. 清晰地命名并标注图中所有关键要素或变量（如质量、力、场、角度等）。

2. 充分描述物理场景或现象（如"质量为 m 的物体沿倾角为 $\theta$ 的有摩擦斜面下滑"、"从点电荷发出的电场线"）。

3. 解释哪些方面对于解决相关物理问题至关重要（如"将重力分解为平行于斜面和垂直于斜面的分量"、"注意库仑定律中的分离距离 r"）。

4. 提供理解该题设为何重要所需的任何背景或上下文。

# E    EXAMPLE PROBLEMS FROM PHYSUNIBENCH

This section would include 5-8 diverse examples from PhysUniBench, showcasing different sub-disciplines, difficulty levels, open-ended vs. MC formats, and problems with diagrams. For each, the problem statement, diagram (if any), and the correct answer/solution would be provided to make the benchmark tangible for the reader.

## Problem 431 Electromagnetism (Open-ended)

**Question (431).**   Two long, straight wires are perpendicular to the plane of the paper and at a distance 0.3m from each other, as shown in the figure. The wires carry currents of $I_1 = 1.9A$ and $I_2 = 5.1A$ in the direction indicated (out of the page). Find the magnitude and direction of the magnetic field (in $\mu T$) at a point A midway between the wires. You need to indicate the direction with a positive or a negative value for the magnetic field. Keep in mind that a vector is positive if directed to the right and negative if directed to the left on the x - axis and it is positive if directed up and negative if directed down on the y - axis. Your answer should be a number with two decimal places, do not include the unit.

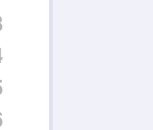

**Difficulty:**   52.

**Correct Answer:**    Step 1: For first wire
Current in first wire: $I_1 = 1.9\,\text{A}$
Distance from point A: $d_1 = \frac{0.3}{2} = 0.15\,\text{m}$
General formula for the magnetic field of a very long current-carrying conductor at a perpendicular distance:

$$B = \frac{\mu_0}{4\pi} \cdot \frac{2I}{d}$$

where $\frac{\mu_0}{4\pi} = 10^{-7}\,\text{T} \cdot \text{m/A}$, $I$ is the current in the wire, and $d$ is the perpendicular distance of the point from the wire.
Magnetic field for the first wire:

$$B_1 = \frac{\mu_0}{4\pi} \cdot \frac{2I_1}{d_1} \quad (+)$$

Substituting values:

$$B_1 = 10^{-7} \cdot \frac{2 \cdot 1.9}{0.15} = 25.33 \times 10^{-7}\,\text{T}$$

Step 2: For second wire
Current in second wire: $I_2 = 5.1\,\text{A}$
Distance from point A: $d_2 = \frac{0.3}{2} = 0.15\,\text{m}$
Magnetic field for the second wire:

$$B_2 = \frac{\mu_0}{4\pi} \cdot \frac{2I_2}{d_2} \quad (-)$$

Substituting values:

$$B_2 = 10^{-7} \cdot \frac{2 \cdot 5.1}{0.15} = -68 \times 10^{-7} \, \text{T}$$

Step 3: Net magnetic field at point A

$$\begin{aligned} B_A &= B_1 + B_2 \\ &= 25.33 \times 10^{-7} + (-68 \times 10^{-7}) \\ &= -42.67 \times 10^{-7} \\ &\approx -4.27 \times 10^{-6} \, \text{T} \end{aligned}$$

Answer:
The net magnetic field at point A is $B_A = 4.27 \times 10^{-6} \, \text{T}$, and it is directed towards the negative $y$-axis.

**Model Answers (Actually wrong):**   Problem:
Two long, straight wires are perpendicular to the plane of the paper and are separated by a distance of $0.3 \, \text{m}$. The currents in the wires are:

$$I_1 = 1.9 \, \text{A}, \quad I_2 = 5.1 \, \text{A}$$

Both currents are directed *out of the page*. Let point $A$ be midway between the wires, i.e., at a distance:

$$r = \frac{0.3}{2} = 0.15 \, \text{m}$$

We are to find the net magnetic field at point $A$, giving direction as: - Positive if the net field is upward (along $+\hat{y}$) - Negative if the net field is downward (along $-\hat{y}$)
Solution:
The magnetic field due to a long straight current-carrying wire at a distance $r$ is:

$$B = \frac{\mu_0 I}{2\pi r}$$

with $\mu_0 = 4\pi \times 10^{-7} \, \text{T} \cdot \text{m/A}$
Thus:

$$B_1 = \frac{\mu_0 I_1}{2\pi r} = \frac{4\pi \times 10^{-7} \cdot 1.9}{2\pi \cdot 0.15} = \frac{2 \times 10^{-7} \cdot 1.9}{0.15} = 2.53 \, \mu\text{T}$$

$$B_2 = \frac{\mu_0 I_2}{2\pi r} = \frac{4\pi \times 10^{-7} \cdot 5.1}{2\pi \cdot 0.15} = \frac{2 \times 10^{-7} \cdot 5.1}{0.15} = 6.80 \, \mu\text{T}$$

Now apply the right-hand rule: - For $I_1$ (out of the page), magnetic field at point $A$ is directed *downward* - For $I_2$ (out of the page), magnetic field at point $A$ is directed *upward*
So the net field at point $A$ is:

$$B_{\text{net}} = B_2 - B_1 = 6.80 - 2.53 = \boxed{4.27 \, \mu\text{T}}$$

Since it's upward (toward $+\hat{y}$), the answer is:

$$\boxed{+4.27}$$

(**Incorrect.** ✗ )

## Problem 1808 Relativity Physics (Multiple-choice)

**Question (1808).**   如图所示，建在SLAC的直线加速器能产生电子和正电子束用于对撞实验，在实验室中电子能量为50GeV。每束包含 $10^{10}$ 个粒子，并且可看作在实验室中半径为 $1.0\mu m$, 长度为 $2.0mm$ 的均匀带电圆柱。画图表示在实验室中测量两粒

子束重叠时的弯转半径 $r$ 与磁场强度 $B$ 的关系。当弯转半径 $r$ 为 $1.0\mu\mathrm{m}$ 时， $B$ 的值是多少？

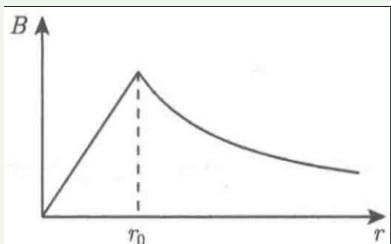

**Choices**
A. 96T.
B. $1.67 \times 10^7$T.
C. 2.7GeV/cm.
D. $8.6 \times 10^4$T.

**Difficulty:** 2.

**Correct Choice:** A.

**Solution** 考虑 $e^+$，设粒子束的长度、半径、粒子数目及电荷密度分别为 $l, r_0, N, \rho$，则：

$$\rho = \frac{eN}{\pi r_0^2 l} \tag{1}$$

正负电子带有相反的电荷，运动方向相反，所以总电流密度为 $J = 2\rho\beta c$。

$$\gamma = \frac{E}{mc^2} = \frac{1}{\sqrt{1-\beta^2}} \tag{2}$$

利用安培环路定理：

$$\oint B \cdot \mathrm{d}l = \mu_0 I = \mu_0 J \pi r_0^2 \tag{3}$$

对于 $r > r_0$，有：

$$2\pi r B = \mu_0 \cdot \frac{2eN}{\pi r_0^2 l} \cdot \beta c \cdot \pi r_0^2 \tag{4}$$

$$B = \mu_0 \cdot \frac{2eN}{2\pi l} \cdot \beta c \cdot \frac{1}{r} = \mu_0 \cdot \frac{eN}{\pi l} \cdot \frac{\beta c}{r} \tag{5}$$

对于 $r < r_0$，有：

$$2\pi r B = \mu_0 \cdot \frac{2eN}{\pi r_0^2 l} \cdot \beta c \cdot \pi r^2 \tag{6}$$

$$B = \mu_0 \cdot \frac{2eN}{l r_0^2} \cdot \beta c \cdot \frac{r}{2} \tag{7}$$

题图显示了 $B$ 随 $r$ 的变化曲线。
当 $r = r_0 = 1.0\,\mu\mathrm{m}$，有：

$$B = \frac{4\pi \times 10^{-7} \times 1.6 \times 10^{-19} \times 10^{10}}{\pi \times 2 \times 10^{-3} \times 10^{-6}} \times 1.0 \times 3.0 \times 10^{8} \approx 96\,\text{T} \qquad (8)$$

**Model Answers (Actually correct):** 题目分析：

SLAC 的直线加速器产生的电子和正电子束可视为半径为 $r_0 = 1.0\,\mu\text{m} = 1.0 \times 10^{-6}\,\text{m}$、长度为 $\ell = 2.0\,\text{mm} = 2.0 \times 10^{-3}\,\text{m}$ 的均匀带电圆柱。每束粒子数为：

$$N = 10^{10}, \quad \text{电子能量} = 50\,\text{GeV}$$

我们要找的是当粒子束重叠、且弯转半径 $r = r_0$ 时的磁场强度 $B$。

第一步：计算束流所产生的电流

总电荷为：

$$q = N \cdot e = 10^{10} \cdot 1.6 \times 10^{-19}\,\text{C} = 1.6 \times 10^{-9}\,\text{C}$$

粒子束以近似光速 $v \approx c = 3.0 \times 10^{8}\,\text{m/s}$ 运动，通过某点的时间为：

$$\Delta t = \frac{\ell}{v} = \frac{2.0 \times 10^{-3}}{3.0 \times 10^{8}} = 6.67 \times 10^{-12}\,\text{s}$$

因此，等效电流为：

$$I = \frac{q}{\Delta t} = \frac{1.6 \times 10^{-9}}{6.67 \times 10^{-12}} \approx 2.4 \times 10^{2}\,\text{A}$$

第二步：计算半径为 $r_0$ 处的磁场强度

对均匀带电圆柱，使用安培环路定理可得半径 $r \le r_0$ 处的磁场为：

$$B = \frac{\mu_0 I r}{2\pi r_0^2}$$

当 $r = r_0$ 时，磁场达到最大值：

$$B = \frac{\mu_0 I}{2\pi r_0}$$

代入数据：

$$\mu_0 = 4\pi \times 10^{-7}\,\text{T} \cdot \text{m/A}, \quad I = 2.4 \times 10^{2}\,\text{A}, \quad r_0 = 1.0 \times 10^{-6}\,\text{m}$$

$$B = \frac{4\pi \times 10^{-7} \cdot 2.4 \times 10^{2}}{2\pi \cdot 1.0 \times 10^{-6}} = \frac{9.6 \times 10^{-5}}{2 \times 10^{-6}} = 48\,\text{T}$$

由于电子束和正电子束重叠，磁场方向相同（同向叠加），因此总磁场为：

$$B_{\text{total}} = 2 \cdot 48\,\text{T} = \boxed{96\,\text{T}}$$

最终答案：

$$\boxed{\text{A. }96\,\text{T}}$$

**Choose A.** (Correct. ✓ )

## Problem 779 Solid State Physics (Open-ended)

**Question (779).** Using the prism, answer the following questions. Write $overrightarrow EC$ in terms of $\vec{a}, \vec{b}$, and $\vec{c}$. Explain. Write $\overrightarrow{CE}$ in terms of $\vec{a}, \vec{b}$, and $\vec{c}$. Explain. What type of vectors are $\overrightarrow{EC}$ and $\overrightarrow{CE}$ in relation to each other?

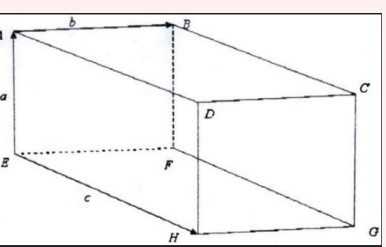

**Difficulty:** 2.

**Correct Answer:** Step 1
Given:

- Prism with edges: $\vec{a}, \vec{b}, \vec{c}$

Objective:

- Express the vector $\vec{EC}$ in terms of $\vec{a}$, $\vec{b}$, and $\vec{c}$.
- Express the vector $\vec{CE}$ in terms of $\vec{a}$, $\vec{b}$, and $\vec{c}$.
- Determine the relationship between the vectors $\vec{EC}$ and $\vec{CE}$.

Explanation:
According to the triangle law of vector addition: when two vectors are represented by two sides of a triangle in sequence (both in magnitude and direction), the third side represents the resultant vector.

Step 2
(a) Writing $\vec{EC}$ in terms of $\vec{a}$, $\vec{b}$, and $\vec{c}$:
To express $\vec{EC}$, we trace the path from point $E$ to $C$ using the prism's edges:

- From $E$ to $F$: vector $\vec{a}$
- From $F$ to $G$: vector $\vec{b}$
- From $G$ to $C$: vector $\vec{c}$

Thus,
$$\vec{EC} = \vec{a} + \vec{b} + \vec{c}$$
(b) Writing $\vec{CE}$ in terms of $\vec{a}$, $\vec{b}$, and $\vec{c}$:
The vector $\vec{CE}$ points from $C$ to $E$, which is the opposite direction of $\vec{EC}$. Therefore:
$$\vec{CE} = -(\vec{a} + \vec{b} + \vec{c})$$

(c) Relationship between $\vec{EC}$ and $\vec{CE}$:
The vectors $\vec{EC}$ and $\vec{CE}$ are opposites:
$$\vec{CE} = -\vec{EC}$$

They have the same magnitude but point in opposite directions, which reflects the vector properties of direction and displacement.

Answer:

(a) $\vec{EC} = \vec{a} + \vec{b} + \vec{c}$

(b) $\vec{CE} = -(\vec{a} + \vec{b} + \vec{c})$

(c) $\vec{EC}$ and $\vec{CE}$ are opposite vectors.

**Model Answers (Paritially correct):** **Given:** The prism is defined such that: - Vector $\vec{a} = \overrightarrow{AE}$ (vertical direction), - Vector $\vec{b} = \overrightarrow{AB}$ (horizontal, width direction), - Vector $\vec{c} = \overrightarrow{EF}$ (depth direction).

We are to express $\overrightarrow{EC}, \overrightarrow{CE}$, and describe the relationship between them.

(a) Express $\overrightarrow{EC}$ in terms of $\vec{a}, \vec{b}, \vec{c}$: To go from point $E$ to point $C$, we can follow the path:

$$E \to A \to B \to C$$

Using vectors:

$$\overrightarrow{EA} = -\vec{a}, \quad \overrightarrow{AB} = \vec{b}, \quad \overrightarrow{BC} = \vec{c}$$

So:

$$\overrightarrow{EC} = \overrightarrow{EA} + \overrightarrow{AB} + \overrightarrow{BC} = -\vec{a} + \vec{b} + \vec{c}$$

$$\boxed{\overrightarrow{EC} = -\vec{a} + \vec{b} + \vec{c}}$$

**(Incorrect.)**

Explanation: We decomposed the vector $\overrightarrow{EC}$ into smaller segments along the known vectors $\vec{a}, \vec{b}, \vec{c}$.

(b) Express $\overrightarrow{CE}$ in terms of $\vec{a}, \vec{b}, \vec{c}$:

$$\overrightarrow{CE} = -\overrightarrow{EC} = -(-\vec{a} + \vec{b} + \vec{c}) = \vec{a} - \vec{b} - \vec{c}$$

$$\boxed{\overrightarrow{CE} = \vec{a} - \vec{b} - \vec{c}}$$

**(Incorrect.)** Explanation: Vector $\overrightarrow{CE}$ is the opposite of $\overrightarrow{EC}$, so we simply negate all components.

(c) What type of vectors are $\overrightarrow{EC}$ and $\overrightarrow{CE}$ with respect to each other? Since:

$$\overrightarrow{CE} = -\overrightarrow{EC}$$

they are:

$$\boxed{\text{opposite vectors (same magnitude, opposite direction)}}$$

Explanation: Opposite vectors have equal magnitude but point in opposite directions. **(Correct. ✓ )**

## Problem 293 Optics (Multiple-choice)

**Question (293).** A right triangle made of a material with a refractive index of 1.5 is immersed in air as shown in the figure. If a beam enters from the left at a distance $y$ from the base, find the distance $x$ at which the refracted beam after the triangle crosses the line extending from the base of the rectangle. The marked angle is $60°$

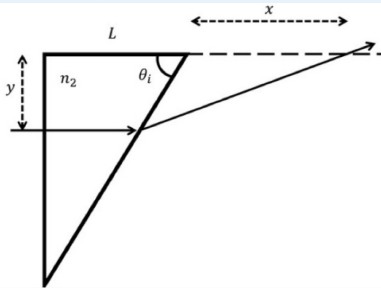

**Choices**

A. The distance is $x = y\sqrt{3}$.
B. The distance is $x = 2.4y$
C. The distance is $x = y \times \tan(60°)$
D. The distance is $x = \frac{y}{\sqrt{3}}$

**Difficulty:** 1.

**Correct Choice:** B.

**Solution**

SOLUTION

The diagram is illustrated below.
In the diagram, a ray of light is incident on the interface between two media forming a rectangular-like structure. The refractive index of the glass-like medium is $\mu_g = 1.5$, and the refractive index of air is $\mu_a = 1$. The angle of incidence is $i = 30°$.

STEP 1: APPLYING SNELL'S LAW

Using Snell's Law:
$$\mu_g \sin i = \mu_a \sin r$$

Substituting the known values:
$$1.5 \cdot \sin 30° = 1 \cdot \sin r$$

Solving for $r$:
$$r = \sin^{-1}(1.5 \cdot \sin 30°) = \sin^{-1}(0.75) \approx 48.6°$$

Then:
$$\angle BOC = 90° - r = 90° - 48.6° = 41.4°$$
$$\angle AOC = i + \angle BOC = 30° + 41.4° = 71.4°$$

Explanation: According to Snell's law, for a specific pair of media and a given light color, the ratio of the sine of the angle of incidence to the sine of the angle of refraction remains constant:
$$\frac{\sin i}{\sin r} = \frac{\mu_a}{\mu_g}$$

STEP 2: USING RIGHT-ANGLE TRIANGLE TRIGONOMETRY

From triangle $\triangle ABO$:
$$\frac{AB}{y} = \tan 30° \Rightarrow AB = y \cdot \tan 30°$$

From triangle $\triangle ACO$:
$$\frac{AC}{y} = \tan 71.4° \Rightarrow AC = y \cdot \tan 71.4°$$

Since $x = BC = AC - AB$, we get:
$$x = y \cdot \tan 71.4° - y \cdot \tan 30° = 2.4y$$

Explanation: In a right-angled triangle, the tangent of an angle is given by:
$$\tan \theta = \frac{\text{opposite}}{\text{adjacent}}$$

ANSWER

The distance at which the refracted beam intersects the line extending from the base of the rectangle is:

$$x = 2.4y$$

Choose B.

**Model Answers (Actually incorrect):**

Given:

A right triangle made of material with refractive index $n_2 = 1.5$ is immersed in air ($n_1 = 1$). A light ray enters from the left at a vertical height $y$ from the base. The top angle of the triangle is marked as $\theta_i = 60°$. We are to find the horizontal distance $x$ from the right side of the triangle to the point where the refracted beam (after exiting) intersects the extension of the base.

Step 1: Refraction at exit face

When light exits the triangle into air, it bends away from the normal (because it goes from denser to rarer medium). By Snell's Law:

$$n_2 \sin(\theta_i) = n_1 \sin(\theta_r) \Rightarrow 1.5 \sin(60°) = \sin(\theta_r) \Rightarrow \sin(\theta_r) = 1.5 \cdot \frac{\sqrt{3}}{2} = \frac{3\sqrt{3}}{4} \approx 1.299$$

Since $\sin(\theta_r) > 1$, total internal reflection occurs. Therefore, instead of exiting, the beam reflects and exits at the bottom horizontal side of the triangle.

But the diagram shows the beam exiting! This implies that the beam is exiting at the top face and the triangle is a 30-60-90 triangle, and we are supposed to *ignore* refraction effects (or the question assumes light continues in same direction as within triangle). So we now treat this as a geometry/trig problem.

Step 2: Use triangle geometry

From the figure, after traveling along the hypotenuse $L$, the light exits and travels along a line making angle $\theta_i = 60°$ with the horizontal.

We want the horizontal distance $x$ from the exit point to where the ray intersects the extension of the base line.

We can use trigonometry:

$$\tan(\theta_i) = \frac{y}{x} \Rightarrow x = \frac{y}{\tan(60°)} = \frac{y}{\sqrt{3}}$$

**Final Answer:**

$$\boxed{\text{D. The distance is } x = \frac{y}{\sqrt{3}}}$$

**Choose D.** (Incorrect. ✗ )

## Problem 25 Thermodynamics (Multiple-choice)

**Question (25).** Two thin metallic bars, one made of zinc and the other made of iron, whose lengths at 300 K are respectively 5.00 m and 12.0 m, are such that the zinc bar is over the iron bar with their left - ends attached to each other with a screw while their other right - ends remain free, as shown in the figure. The linear expansion coefficients of zinc and iron are respectively $\alpha_{zinc} = 3.00 \times 10^{-5} K^{-1}$ and $\alpha_{iron} = 1.00 \times 10^{-5} K^{-1}$. Neglecting the thickness of the bars, determine: a) The change in the distance between the ends A and B when the bars are heated to the temperature of 400 K; b) The distance to point A from a point C on the zinc bar such that the length from C to A remains constant during the heating of the bars.

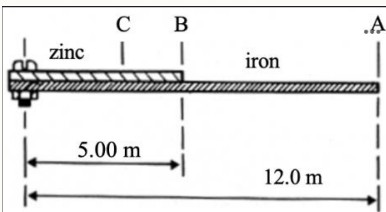

**Choices**

A. The change in distance between A and B is 0.3 cm; the distance from A to C is 11 m.
B. The change in distance between A and B is 0.003 m; the distance from A to C is 1.25 m.
C. The change in distance between A and B is 0.3 cm; the distance from A to C is 11 m.
D. The change in distance between A and B is 0.015 m; the distance from A to C is 9 m.

**Difficulty:** 5.

**Correct Choice:** C.

**Solution** (a) Explanation: Since the temperature is being increased, due to the phenomenon of thermal expansion, the length of both the zinc and iron bars will increase. The increase in length $\Delta L$ of a bar is given by the relation:

$$\Delta L = \alpha L (T_f - T_i) \qquad (1)$$

where $\alpha$ is the coefficient of linear expansion, $L$ is the original length, and $T_i$, $T_f$ are the initial and final temperatures, respectively.
For Zinc Bar: Substitute $\alpha_{\text{zinc}} = 3 \times 10^{-5}\,\text{K}^{-1}$, $L = 5\,\text{m}$, $T_i = 300\,\text{K}$, $T_f = 400\,\text{K}$ into equation (1):

$$\Delta L_{\text{zinc}} = 3 \times 10^{-5}\,\text{K}^{-1} \times 5\,\text{m} \times (400\,\text{K} - 300\,\text{K})$$
$$= 3 \times 10^{-5} \times 5 \times 100\,\text{m}$$
$$= 0.015\,\text{m}$$

For Iron Bar: Substitute $\alpha_{\text{iron}} = 1 \times 10^{-5}\,\text{K}^{-1}$, $L = 12\,\text{m}$, $T_i = 300\,\text{K}$, $T_f = 400\,\text{K}$:

$$\Delta L_{\text{iron}} = 1 \times 10^{-5}\,\text{K}^{-1} \times 12\,\text{m} \times (400\,\text{K} - 300\,\text{K})$$
$$= 1 \times 10^{-5} \times 12 \times 100\,\text{m}$$
$$= 0.012\,\text{m}$$

The change in distance between $A$ and $B$ is:

$$\Delta L_{AB} = \Delta L_{\text{zinc}} - \Delta L_{\text{iron}}$$
$$= 0.015\,\text{m} - 0.012\,\text{m}$$
$$= 0.003\,\text{m}$$
$$= 0.3\,\text{cm}$$

So, the change in distance between $A$ and $B$ is $\Delta L_{AB} = 0.3\,\text{cm}$.
(b) Let the distance of point $C$ from the left end be $d$. Then the remaining length of the zinc bar is:

$$BC = 5 - d$$

Explanation: To keep the length constant between points $A$ and $C$, the expansion in both bars must be equal:

$$\Delta L_{\text{zinc,AC}} = \Delta L_{\text{iron}} \qquad (2)$$

Expansion in the zinc bar is given by:

$$\Delta L_{\text{zinc,AC}} = \alpha_{\text{zinc}}(5 - d)(T_f - T_i) \qquad (3)$$

Equating (2) and (3):

$$\alpha_{\text{zinc}}(5 - d)(T_f - T_i) = \Delta L_{\text{iron}}$$

Substitute known values:

$$3 \times 10^{-5} \times (5 - d) \times 100 = 0.012$$
$$(5 - d) \times 3 \times 10^{-3} = 0.012$$
$$5 - d = \frac{0.012}{3 \times 10^{-3}}$$
$$d = 5 - \frac{0.012}{3 \times 10^{-3}}$$
$$d = 1\,\text{m}$$

The distance of point $A$ from point $C$ is:

$$s = 12 - d$$
$$s = 12\,\text{m} - 1\,\text{m}$$
$$s = 11\,\text{m}$$

Answer

    (a) The change in distance between $A$ and $B$ is $\Delta L_{AB} = 0.3\,\text{cm}$.

    (b) The distance of point $A$ from point $C$ is $s = 11\,\text{m}$.

Choose C.

**Model Answers (Actually false):**

**Given:**

- Initial temperature: $T_0 = 300\,\text{K}$
- Final temperature: $T_f = 400\,\text{K}$
- Temperature change: $\Delta T = 100\,\text{K}$
- Length of zinc bar: $L_{\text{zinc}} = 5.00\,\text{m}$
- Length of iron bar: $L_{\text{iron}} = 12.0\,\text{m}$
- Linear expansion coefficients:

$$\alpha_{\text{zinc}} = 3.00 \times 10^{-5}\,\text{K}^{-1}, \quad \alpha_{\text{iron}} = 1.00 \times 10^{-5}\,\text{K}^{-1}$$

(a) Change in the distance between points A and B:

$$\Delta L_{\text{zinc}} = \alpha_{\text{zinc}} \cdot L_{\text{zinc}} \cdot \Delta T = (3.00 \times 10^{-5}) \cdot 5.00 \cdot 100 = 0.015\,\text{m}$$

$$\Delta L_{\text{iron}} = \alpha_{\text{iron}} \cdot L_{\text{iron}} \cdot \Delta T = (1.00 \times 10^{-5}) \cdot 12.0 \cdot 100 = 0.012\,\text{m}$$

$$\Delta d = \Delta L_{\text{zinc}} - \Delta L_{\text{iron}} = 0.015 - 0.012 = 0.003\,\text{m} = \boxed{0.3\,\text{cm}}$$

(b) Distance from point A to a point C on the zinc bar such that the length from C to A remains constant:

Let the distance from the screw to point C be $x$. Then the segment from C to A consists of:

- Zinc segment: $L_{\text{zinc, C}} = 5.00 - x$
- Iron segment: $L_{\text{iron}} = 12.0$

For the total expansion of segment CA to be zero:

$$\alpha_{\text{zinc}}(5.0 - x) + \alpha_{\text{iron}}(12.0) = 0$$

$$(3.00 \times 10^{-5})(5.0 - x) + (1.00 \times 10^{-5})(12.0) = 0$$

$$(1.5 \times 10^{-4} - 3.00 \times 10^{-5}x) + 1.2 \times 10^{-4} = 0$$

$$\Rightarrow \quad 2.7 \times 10^{-4} = 3.00 \times 10^{-5}x$$

$$\Rightarrow \quad x = \frac{2.7 \times 10^{-4}}{3.00 \times 10^{-5}}$$

$$= 9.0$$

So point C is 9.0 m from the screw (on the zinc bar), which is beyond the zinc's total length of 5.0 m. This implies that point C lies $x = 4.0$ m from the screw (i.e., within the zinc bar), and:

$$\text{Distance from C to A} = L_{\text{iron}} + (5.0 - x) = 12.0 + (5.0 - 4.0) = \boxed{13.0 \, \text{m}}$$

Final Answers:

- (a) The change in the distance between A and B is $\boxed{0.003 \, \text{m} \, (0.3 \, \text{cm})}$

- (b) The distance from A to C such that CA remains constant is $\boxed{13.0 \, \text{m}}$

So none of the options are fully correct, but Option A is partially correct, capturing the correct change in distance.
**Choose A.** (Incorrect. ✗ )

## Problem 795 Mechanics (Open-ended)

**Question (795).** Two objects are moving towards each other with velocities $\vec{V}_1$ and $\vec{V}_2$ on a frictionless surface. They collide completely inelastically and move together after the collision with a velocity $|\vec{V}_3| = 16.8 m/s$ as shown below. If the ratio of the masses is $m_1/m_2 = 1.38$ and the angles are $\theta_1 = 59°$ and $\theta_2 = 39°$, what is the magnitude of $\vec{V}_1$ in units of m/s? Express your answer using one decimal place.

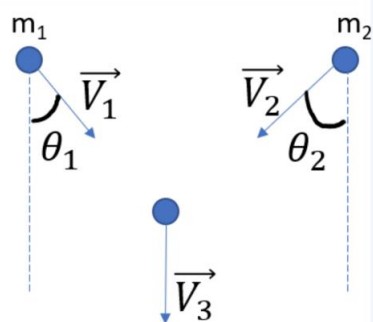

**Difficulty:** 5.

**Correct Answer:**

SOLUTION

Step 1: Conservation of Momentum
In a completely inelastic collision, momentum is conserved in both the $x$- and $y$-directions. Explanation: The total momentum of a closed system remains constant if no external forces act on it. $x$-direction:

$$m_1 V_1 \cos\theta_1 + m_2 V_2 \cos\theta_2 = (m_1 + m_2)V_3 \tag{1}$$

$y$-direction:

$$m_1 V_1 \sin \theta_1 = m_2 V_2 \sin \theta_2 \qquad (2)$$

Step 2: Relation Between $m_1$ and $m_2$

Given:

$$\frac{m_1}{m_2} = 1.38 \Rightarrow m_1 = 1.38 m_2$$

This allows all terms to be expressed in terms of $m_2$. Substitute Known Values:

- $V_3 = 16.8 \,\text{m/s}$
- $\theta_1 = 59°$
- $\theta_2 = 39°$
- $\frac{m_1}{m_2} = 1.38$

Explanation: Momentum is a measure of an object's motion and is defined as the product of its mass and velocity.

Step 3: Solve for $V_1$

From equation (1), rearranged in terms of $V_2$:

$$V_2 = \frac{-m_1 V_1 \cos \theta_1}{m_2 \cos \theta_2} = \frac{-1.38 \cdot V_1 \cos 59°}{\cos 39°}$$

Now substitute into the $y$-momentum equation:

$$m_1 V_1 \sin \theta_1 + m_2 V_2 \sin \theta_2 = (m_1 + m_2) V_3$$

Substituting $m_1 = 1.38 m_2$:

$$1.38 V_1 \sin 59° + V_2 \sin 39° = (1.38 + 1) \cdot 16.8$$

Substitute for $V_2$:

$$1.38 V_1 \sin 59° - \left( \frac{1.38 \cdot V_1 \cos 59° \cdot \sin 39°}{\cos 39°} \right) = 2.38 \cdot 16.8$$

Factor out $V_1$:

$$V_1 \left[ 1.38 \cdot \sin 59° - \frac{1.38 \cdot \cos 59° \cdot \sin 39°}{\cos 39°} \right] = 39.984$$

Approximate:

$$V_1 (1.18 - 0.58) = 39.984 \Rightarrow V_1 = \frac{39.984}{0.6} = 66.6 \,\text{m/s}$$

Explanation: Vectors have both magnitude and direction. In many physics problems involving angles, breaking vectors into their $x$ and $y$ components using trigonometric functions is essential.

Answer The final velocity of mass $m_1$ is:

$$V_1 = 66.6 \,\text{m/s}$$

**Model Answers (Incorrect):**   Given:

- Final velocity after perfectly inelastic collision: $|\vec{V_3}| = 16.8$ m/s
- Mass ratio: $\frac{m_1}{m_2} = 1.38$
- Angles: $\theta_1 = 59°, \quad \theta_2 = 39°$
- After collision, the bodies stick together and move straight down ($\vec{V_3}$ is vertical)

Step 1: Use conservation of momentum

Since the final motion is vertical, the total horizontal momentum before the collision must be zero.

Let $\vec{V_1}$ and $\vec{V_2}$ be the speeds of $m_1$ and $m_2$, respectively. The horizontal momentum equation:

$$m_1 V_1 \cos \theta_1 = m_2 V_2 \cos \theta_2 \quad (1)$$

The vertical momentum equation:

$$m_1 V_1 \sin\theta_1 + m_2 V_2 \sin\theta_2 = (m_1 + m_2)V_3 \quad (2)$$

Step 2: Eliminate $V_2$ using equation (1)
From (1):

$$V_2 = \frac{m_1}{m_2} \cdot \frac{\cos\theta_1}{\cos\theta_2} V_1$$

Substitute into (2):

$$m_1 V_1 \sin\theta_1 + m_2 \left( \frac{m_1}{m_2} \cdot \frac{\cos\theta_1}{\cos\theta_2} V_1 \right) \sin\theta_2 = (m_1 + m_2)V_3$$

Simplify:

$$m_1 V_1 \left[ \sin\theta_1 + \frac{\cos\theta_1}{\cos\theta_2} \sin\theta_2 \right] = (m_1 + m_2)V_3$$

Let $r = \frac{m_1}{m_2} = 1.38$, so $m_1 = 1.38 m_2$. Then $m_1 + m_2 = (1.38 + 1)m_2 = 2.38 m_2$.
Substituting back:

$$1.38 m_2 V_1 \left[ \sin 59° + \frac{\cos 59°}{\cos 39°} \sin 39° \right] = 2.38 m_2 \cdot 16.8$$

Cancel $m_2$:

$$1.38 V_1 \left[ \sin 59° + \frac{\cos 59°}{\cos 39°} \sin 39° \right] = 2.38 \cdot 16.8$$

Compute values:

$$\sin 59° \approx 0.8572, \quad \cos 59° \approx 0.5150, \quad \sin 39° \approx 0.6293, \quad \cos 39° \approx 0.7771$$

$$1.38 V_1 \left[ 0.8572 + \frac{0.5150}{0.7771} \cdot 0.6293 \right] = 39.984$$

$$\frac{0.5150}{0.7771} \cdot 0.6293 \approx 0.4168 \Rightarrow \text{sum in brackets} \approx 0.8572 + 0.4168 = 1.2740$$

$$1.38 V_1 \cdot 1.2740 = 39.984 \Rightarrow V_1 = \frac{39.984}{1.38 \cdot 1.2740} \approx \frac{39.984}{1.7581} \approx \boxed{22.7 \text{ m/s}}$$

—

**Final Answer:**
$$\boxed{22.7 \text{ m/s}}$$

(**Incorrect.** ✗ )

## Problem 912 Molecular Atomic & Subatomic Physics (Multiple-choice)

**Question (912).** Hydrogen atoms feature energy levels as shown below.Which photon energy will not cause an electron to be excited or ionized in a ground state hydrogen atom?

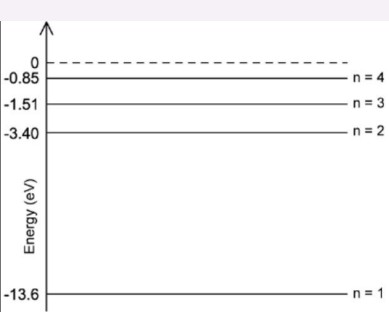

**Choices**

A.  10.2eV.

B.  12.29eV.

C.  12.75eV.

D.  15.0eV.

**Difficulty:**   1.

**Correct Choice:**   B.

**Solution**

SOLUTION

Step 1: Given Information
Hydrogen atoms have discrete energy levels. The energy difference between the first level $(n = 1)$ and higher levels is calculated as follows:

$$E_a = E_2 - E_1 = -3.4\,\text{eV} - (-13.6\,\text{eV}) = 10.2\,\text{eV}$$

Explanation: Ionization energy is the energy required to completely remove an electron from an atom.

Step 2: Additional Energy Transitions

$$E_b = E_3 - E_1 = -1.51\,\text{eV} - (-13.6\,\text{eV}) = 12.09\,\text{eV}$$
$$E_c = E_4 - E_1 = -0.85\,\text{eV} - (-13.6\,\text{eV}) = 12.75\,\text{eV}$$

Ionization Energy:

$$\text{Ionization energy} = 0\,\text{eV} - (-13.6\,\text{eV}) = 13.6\,\text{eV}$$

The given energy of $15\,\text{eV}$ is sufficient to ionize a hydrogen atom.
Answer
Correct Option: (b) 12.29 eV Choose B.

**Model Answers (Actually correct):**
Given: The energy levels for a hydrogen atom are:

$$E_1 = -13.6\,\text{eV}$$
$$E_2 = -3.40\,\text{eV}$$
$$E_3 = -1.51\,\text{eV}$$
$$E_4 = -0.85\,\text{eV}$$

A ground state hydrogen atom has its electron initially in $n = 1$. The electron can be:

- **Excited** to a higher energy level (e.g., from $n = 1$ to $n = 2, 3, 4, \ldots$) if it absorbs a photon with energy equal to the energy difference between levels.

- **Ionized** if it absorbs energy $\geq 13.6\,\text{eV}$, which brings it from $E_1 = -13.6\,\text{eV}$ to $E = 0$.

Allowed excitation transitions from $n = 1$:

$$E_{1 \to 2} = -3.40 - (-13.6) = 10.2 \, \text{eV}$$
$$E_{1 \to 3} = -1.51 - (-13.6) = 12.09 \, \text{eV}$$
$$E_{1 \to 4} = -0.85 - (-13.6) = 12.75 \, \text{eV}$$

Ionization energy:

$$E_{1 \to \infty} = 0 - (-13.6) = 13.6 \, \text{eV}$$

Now examine the options:

- A. $10.2 \, \text{eV}$: matches $1 \to 2$
- B. $12.29 \, \text{eV}$: not matching any allowed transition
- C. $12.75 \, \text{eV}$: matches $1 \to 4$
- D. $15.0 \, \text{eV}$: sufficient for ionization

Correct Answer:

$$\boxed{\text{B. } 12.29 \, \text{eV}}$$

(Correct. ✓ )

## Problem 3297 Quantum Mechanics (Open-ended)

某些中心势的弹性散射可合理地用一级 Born 近似来计算。实验结果给出下列以 $q = |k^{prime} - k|$（动量转移大小）为变量的散射截面曲线，如图中给出的参数回答：在小距离上势 $V$ 的行为如何？

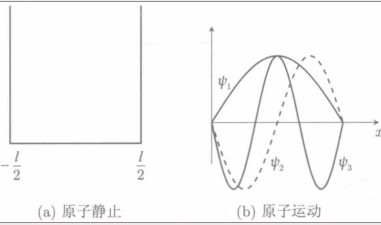

(a) 原子静止    (b) 原子运动

**Difficulty:** 2.

**Correct Choice:** C.

**Correct Answer:** 注意到在大 $q$ 下散射截面的行为，可知此时 Born 积分的主要贡献来自于 $qr \leqslant \pi$ 范围。由于正弦函数的振荡性及其有界性，在此范围之外的积分几乎为零。因此只需对小范围内的 $r$ 进行积分。

设在小距离下，$V(r) \sim r^n$（之所以这样假设，可从下面运算中看出），则有：

$$f(\theta) = -\frac{2m}{\hbar^2} \int \mathrm{d}r \, r^2 V(r) \frac{\sin(qr)}{qr}$$

近似为：

$$\approx -\frac{2m}{\hbar^2} \int_0^x \mathrm{d}(qr) \, (qr)^2 V(qr) \frac{\sin(qr)}{qr} \cdot q^{-(3+n)}$$

化简得：

$$= \frac{1}{q^{3+n}} \left[ -\frac{2m}{\hbar^2} \int_0^\pi \mathrm{d}x \, x^2 V(x) \frac{\sin x}{x} \right]$$

将该结果与散射截面的行为相比较，可得：

$$\frac{N}{2} = 3 + n$$

因此，小距离范围内 $V$ 的行为为：

$$V \sim r^{(N/2-3)}$$

**Model Answers (Correct):** 解答：
在一级 Born 近似下，弹性散射振幅

$$f(\mathbf{q}) \propto \widetilde{V}(\mathbf{q}) = \int e^{i\mathbf{q}\cdot\mathbf{r}} V(\mathbf{r})\, \mathrm{d}^3 r$$

因此

$$\frac{\mathrm{d}\sigma}{\mathrm{d}\Omega} \sim |f(\mathbf{q})|^2 \sim \left|\widetilde{V}(\mathbf{q})\right|^2.$$

题中给出

$$\frac{\mathrm{d}\sigma}{\mathrm{d}\Omega} \sim q^{-N} \quad \Longrightarrow \quad \left|\widetilde{V}(q)\right|^2 \sim q^{-N} \Rightarrow \widetilde{V}(q) \sim q^{-N/2}.$$

而势的傅里叶反变换中，若 $\widetilde{V}(q) \sim q^{-s}$，则

$$V(r) \sim r^{s-3}.$$

取 $s = N/2$，得

$$\boxed{V(r) \sim r^{(N/2)-3}}$$

即在小距离（对应大 $q$）处，中心势 $V(r)$ 呈幂律行为 $V(r) \propto r^{\frac{N}{2}-3}$。
(Correct. ✓ )

## F  DIFFICULTY AND SUB-DISCIPLINES EVALUATION

To delineate capability boundaries and identify characteristic failure modes across physics domains, we report difficulty-stratified results within each sub-discipline; for illustration, figures present accuracy (%) of selected models on Mechanics and Electromagnetism as a function of increasing difficulty.

This expanded analysis yields two principal findings. First, performance declines consistently with rising reasoning complexity across all models, corroborating the validity of our difficulty stratification. Second, the magnitude of the difficulty-induced drop varies by sub-discipline; it is sharper in Mechanics than in Electromagnetism, indicating nonuniform gaps in physical reasoning skills.

Table 9: Performance comparison across different models on Mechanics and Electromagnetism tasks by difficulty levels (%).

| Model | D1 | D2 | D3 | D4 | D5 |
|---|---|---|---|---|---|
| *Mechanics* | | | | | |
| **GPT-5** | **72.8** | **59.7** | **54.4** | **44.9** | **29.4** |
| GPT-4o | 69.1 | 37.7 | 22.8 | 20.3 | 21.6 |
| GPT-4o-mini | 49.4 | 28.6 | 28.1 | 13.0 | 19.6 |
| Claude-3.5-Sonnet | 50.6 | 36.4 | 12.3 | 24.6 | 15.7 |
| Gemini2.5-Pro | 65.4 | 42.9 | 40.4 | 31.9 | 19.6 |
| Qwen2.5-VL-72B | 70.4 | 45.5 | 21.1 | 18.8 | 13.7 |
| *Electromagnetism* | | | | | |
| **GPT-5** | **86.5** | **57.9** | **45.8** | **45.7** | **43.9** |
| GPT-4o | 78.4 | 50.0 | 27.1 | 30.4 | 17.5 |
| GPT-4o-mini | 62.2 | 39.5 | 18.6 | 15.2 | 14.0 |
| Claude-3.5-Sonnet | 64.9 | 36.8 | 25.4 | 19.6 | 10.5 |
| Gemini2.5-Pro | 67.6 | 50.0 | 33.9 | 26.1 | 29.8 |
| Qwen2.5-VL-72B | 67.6 | **57.9** | 32.2 | 32.6 | 12.3 |

## G    DATA INTEGRITY COMPARED RELATED BENCHMARKS

We assess potential overlap with existing physics benchmarks via near-duplicate detection based on question-embedding similarity, using SemHash (van Dongen & Tulkens, 2025) with Potion-Multilingual-128M as the embedding model. The intersection is minimal: 3 similar entries out of 3,000 for PhysX (Shen et al., 2025a) and 4 similar entries out of 1,200 for PhysReason (Zhang et al., 2025) under the 0.9 default similarity threshold. Moreover, UGPhysics (Xu et al., 2025b) is text-only and thus misaligned with the multimodal design of our benchmark, so it is not included in systematic comparison.

## H    EXAMPLES OF SHORT, MEDIUM, AND LONG CAPTIONS

---

### Electric Field Problem Example

**Original Question:**
Two point charges are positioned at opposite corners of a rectangle. A charge $q_1 = +1\,\text{nC}$ is located at the bottom-left corner, and a charge $q_2 = -2\,\text{nC}$ is located at the top-right corner. The rectangle has dimensions of $2\,\text{cm}$ (horizontal) by $1\,\text{cm}$ (vertical). Calculate the electric field at the top-left corner of the rectangle.

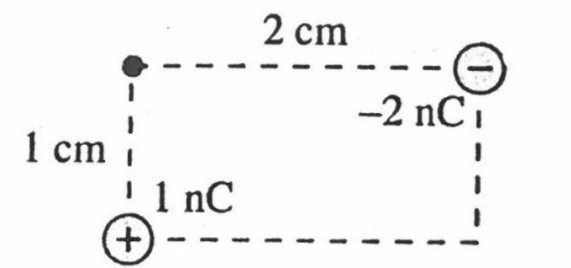

**Long Caption:**
This figure shows a rectangular arrangement of two point charges and a field evaluation point. A positive charge $q_1 = +1\,\text{nC}$ sits at the bottom-left corner, and a negative charge $q_2 = -2\,\text{nC}$ sits at the top-right corner. The black dot indicates the point at which the electric field is to be calculated; it lies at the top-left corner of the rectangle.
The geometry is as follows: the vertical separation between $q_1$ and the dot is $r_1 = 1\,\text{cm}$, and the horizontal separation between $q_2$ and the dot is $r_2 = 2\,\text{cm}$. These distances determine the magnitudes of the individual electric field contributions through the relation $E = k|q|/r^2$, where $k = 9 \times 10^9\,\text{N} \cdot \text{m}^2/\text{C}^2$.
Because electric fields from positive charges point away from the source, the field produced by $q_1$ at the dot is directed upward. The field produced by the negative charge $q_2$ points toward that charge and therefore lies horizontally to the right. Computing the net field requires vector addition of these perpendicular contributions, making the geometry and charge signs essential elements of the setup.

**Medium Caption:**
The image shows a rectangular setup with a $+1\,\text{nC}$ charge at the bottom left and a $-2\,\text{nC}$ charge at the top right, with dimensions of $2\,\text{cm}$ by $1\,\text{cm}$. The point of interest is at the top left corner, where the electric field is calculated using Coulomb's law by determining the vector sum of the fields from both charges. This scenario highlights the importance of charge magnitude, distance, and vector addition in electrostatics.

**Short Caption:**

---

The image shows a rectangular setup with a $+1\,\mathrm{nC}$ charge at the bottom left and a $-2\,\mathrm{nC}$ charge at the top right, focusing on the electric field at the top left corner.

## I  THE USE OF LARGE LANGUAGE MODELS

Large language models (LLMs) were used solely as auxiliary tools to assist grammar checking, and formatting adjustments.

