# OpenReview forum: "PhysUniBench: A Multi-Modal Physics Reasoning Benchmark at Undergraduate Level"
_ICLR.cc/2026/Conference — Submitted to ICLR 2026_

### Official Review · Reviewer_ax6M · 2025-10-15

**Soundness:** 1
**Presentation:** 2
**Contribution:** 2
**Rating:** 2
**Confidence:** 5

**Summary:**

This paper introduces PhysUniBench, a large-scale multimodal benchmark specifically designed to evaluate AI models on undergraduate-level physics problems. The benchmark contains 3,304 carefully curated physics questions spanning 8 major sub-disciplines (Classical Mechanics, Electromagnetism, Optics, Quantum Mechanics, Thermodynamics, Solid Physics, Relativity Physics, and Molecular/Atomic/Subatomic Physics). The benchmark has EN and CN language coverage and has MC as well as Open-Ended QA (OE) in terms of question format, stratified across 5 difficulty level and evaluated using a variety of frontier (M)LLM models.

**Strengths:**

1. Multimodal:  It's nice that each question has exactly one diagram, (altho it also raises the question as to sometimes in the real use case of AI4Science the ability to interpret multiple images and cross-ref between them could be a valuable capability to probe)
2. Difficulty Rating: The authors wrote that each problem is annotated with a fine-grained difficulty level ranging from 1 to 5, which then in turn shows a corresponding performance gradient in the evaluation phase (harder problems = lower scores) which is good for testing MLLMs on a difficulty gradient. (Note: I later realized the difficulty rating was also based on LLM performance, which may be considered circular reasoning and could be strengthened by e.g. human-AI cross-validation and other checks for construct validity)
3. I also noticed that the stratified difficulty has roughly similar number of questions around 660, which is a nice property for statistical analysis in general when one wants to test the model performance across different difficulty level without having to worry about sample size as a confounder

**Weaknesses:**

Disclaimer: I come from a mix of CS+Physics background and have at least graduate student level of physics knowledge, so I believe I am qualified to speak to the validity to an undergraduate level physics benchmark.

APS: American Physical Society (which is generally regarded as one of the most credible non-profit org on physics research)

1. Trivial mistakes in the Physics discipline: I found some of the mistakes made in this paper to be quite trivial:

For example:"Solid Physics" is not the correct term in physics, but it should be "Solid State Physics" or "Solid-State Physics"
I encourage everyone to simply do a Google Search of the word "Solid Physics" you will find everything is about "Solid-State Physics" because "Solid Physics" is not really a term we use in Physics.

[Start quote from Wiki] Solid-state physics is the study of rigid matter, or solids, through methods such as solid-state chemistry, quantum mechanics, crystallography, electromagnetism, and metallurgy. It is the largest branch of condensed matter physics. [End quote]

(This is not some proprietary knowledge, but a simple Wikipedia search would yield this explanation/definition as to what it means and what the correct term is)

2. Therefore, I respectfully question if there were indeed "expert curation" going on by the expert of physics in constructing such a benchmark, how could these "experts" have not noticed such a basic mistake in the terminology? This gave me no choice but to question the validity of the claimed expert construction process, because it's hard to believe that if any of the annotation was actually done by experts, or even inspected/guided by experts, they would not notice this basic mistake.

3. Very similarly to the point I raised above,"Relativity Physics" is also not really a term we often use in Physics, because the Theory of Relativity has both "Special Relativity"(which applies to all physical phenomena in the absence of gravity.) as well as "General Relativity" (which explains the law of gravitation and its relation to other laws of physics) where GR is often classified into Gravitation, Cosmology and Astrophysics (this term is stipulated by APS). So it's also quite confusing to me as to why the authors choose to specifically list "Relativity Physics" in parallel on the same level of e.g. "Quantum Mechanics" because in Physics they are definitely not classified to be on the same level. (You can e.g. check PhySH published by the APS)

4. Apart from trivial mistakes in Physics, the coverage of this benchmark has a couple of flaws:

a. The domain distribution is quite uneven, where EM and CM constitutes nearly half and the rest are much smaller

b. There are domains which are totally not covered like Astrophysics and Cosmology, Quantum Optics, Quantum Info etc. depending on the curriculum you choose to follow I can often see some of these classes being offered to undergrad in many university-level institutions across various countries, especially to physics major in their final year(s).

5. Many of the details aren't super clear to me: For example, if we are trusting LLM with stratification and LLM-as-a-judge for OE questions, are there any validity check by experts to see if they actually do a good job? There should be a inter-annotator or human-AI cross-validation so as to see if there's any false positive/negatives and report the statistics accordingly

6. Also, there lacks a human baseline by different level of human candidates, or at least undergraduates?

7. For difficulty stratification: If you are relying on MLLM 16-rollout success rate to classify the questions, aren't you kind of circular reasoning here in terms of:

a. Use model accuracy to classify the questions from 0 to 5

b. Then say ok this classification is good because models indeed perform correspondingly, but that's actually by design, right? since it is indeed the standard by which you partition the questions in the very first place, so this point in discussion seems like circular reasoning to me. Granted you could say the partition is done by a single model and the eval on many models, but the pretraining corpora of these models are likely overlapping a lot, so I don't think this classification makes much sense to me as to how it's designed.

8. The related work coverage and comparison can be more comprehensive, some general benchmarks like ARB has a physics portion and more dedicated benchmarks like HiPhO and SeePhys and Multi-Physics should be more comprehensively compared in this paper

9. Also, the evaluated models could be more comprehensive (only 2 text-based LLM, maybe for multimodal benchmark you don't really need that? Or if you want to compare LLM against MLLM you should have roughly similar number of models evaluated on each side?) and more thorough error analysis could significantly strengthen this work.

Therefore, I believe this work has significant improvement to be done that could better contribute to the community at large, I encourage the authors to take some more time for improvement. And while we appreciate the author's good effort in constructing this benchmark, it is essential to remember for all of us that AI4Science benchmark (not just in physics, but any other domains as well) should have some input from domain experts in respective field to ensure their construct validity.

**Questions:**

I've mostly put the questions inside the last section of weakness. Some additional ones

1. For evaluated models: Even for MLLMs, it's mostly GPT models, also o3 and o4-mini are o-series not GPT-series, why isn't Claude/Gemini/Grok/other series as comprehensively tested as GPT/OpenAI models? It would probably be nice to test scaling and time evolution of many families of models, or different size models of the same open-sourced families? In general I think the evaluation and discussion section can be improved significantly;

2. For the OE judge logs can you show some examples of model output against LLM judge verdict, so that we can see how exactly they perform in terms of judging the output? I fear rubric-based grading may not be very optimal given GPT-4o's limited capability.

3. Maybe the authors can better present the novelty of this work other than "multi-phase construction by advanced AI models" because one could always just use stronger models to get a much more difficult benchmarks in that case (e.g. replace Qwen-VL with Gemini-2.5-Pro or GPT-5-High?) then the benchmark would be much less prone to saturation over time comparing to the current version?

4. Also, can you better articulate how OE questions are judged? The current Figure 1 has some of them just "Steps" and then the answer is contained in the last step, but some others has a dedicated "Answer"? Are OE questions judged based on "Answer" entry and not on the intermediate process? If on the intermediate process maybe you could shed some light (e.g. by giving an example in the main text, to show this grading, which would imo strengthen this work a lot)

5. As previously mentioned could you better organize the appendix with a Table of Contents? It takes a lot of time for reviewers to read thru >20 pages of appendix without a clear structure.

6. Although this may not warrant an ethics review, how exactly have the authors complied with licensing/data protection laws in the process of curating this benchmark? This may warrant a clearer presentation in Section 3. Currently most of the descriptions are really vague and high-level only.

---

> ### Author Response · Authors · 2025-11-25
> **First Response (1), Regarding Reviewer's (ax6M) Comments**
>
> **Weakness-1: Trivial mistakes in Physics discipline. "Solid Physics" is not the correct term in physics, but it should be "Solid State Physics" or "Solid-State Physics".**
>
> We thank the reviewer for carefully pointing this out. We agree that “Solid-State Physics” is the widely accepted term in the physics community, while “Solid Physics” is less formal and potentially misleading. This was purely a wording oversight on our side, and we appreciate the reviewer for highlighting it.
> We will modsify all instances of “Solid Physics” to the proper term “Solid-State Physics” throughout the paper and supplementary materials. This wording update does not affect our results or analysis. We appreciate the reviewer’s attention to detail.
>
> **Weakness-2: Question if there were indeed "expert curation" going on by experts in physics in constructing such a benchmark.**
>
> The terminology issue (“Solid Physics”) was an editorial oversight made during manuscript preparation by a non-native English speaker, rather than an error in the expert-curated physics content itself. Our domain experts were asked to verify the scientific correctness and validity of the provided questions and solutions and were not responsible for meta-information such as category labels, which led to this terminology oversight during manuscript preparation.
> To clarify:
> - Physics questions, solutions and LLM verification process were reviewed and verified by experts.
> - The terminology issue affected only one category name, not the technical correctness of the benchmark and the validity of the expert curation process.
> - We will correct it to “Solid-State Physics” throughout the manuscript to improve the clarity and professionalism.
>
> **Weakness-3: Relativity Physics" is also not really a term we often use in Physics.**
>
> We agree that “Special Relativity” and “General Relativity” are the standard disciplinary labels as sugged by PhySH. In our benchmark, the grouping of relativity-focused questions was designed for undergraduate-level evaluation, and follows the structure used in the well-known undergraduate-level Chinese textbook "A Grand Dictionary of Physics Problems and Solutions", which organizes exam questions from the CUSPEA project. In that reference, relativity topics are presented as a standalone category, which we adopted for practical clarity in benchmarking.
> To improve disciplinary precision and avoid confusion, we will clarify the inclusion of relativity physics. This refinement concerns only naming conventions and does not impact benchmark content, expert review, or evaluation validity. We appreciate the reviewer’s attention to accurate terminology.

---

> ### Author Response · Authors · 2025-11-25
> **First Response (2), Regarding Reviewer's (ax6M) Comments**
>
> **Weakness-4-a: The domain distribution is quite uneven, where EM and CM constitute nearly half and the rest are much smaller.**
>
> The current distribution reflects the realistic weighting of undergraduate physics curricula, where Classical Mechanics (CM) and Electromagnetism (EM) constitute the majority of foundational training and exam questions.
> At the same time, we agree that expanding other sub-disciplines (e.g., Relativity, Quantum, Thermal) would further enrich the benchmark. Our current scale is primarily constrained by (1) expert curation cost and (2) the scarcity of high-quality, open-licensed question sources in those domains. We also think that expanding the coverage would be valuable, and we plan to pursue this in future updates when resources allow.
>
> **Weakness-4-b: There are domains which are totally not covered.**
>
> We agree that these subjects are important parts of certain upper-level undergraduate physics curricula. For PhysUniBench, we intentionally focused on the core foundational subjects that are required across nearly all physics programs. Our current sub-discipline structure follows the widely used undergraduate reference "A Grand Dictionary of Physics Problems and Solutions", which reflects the dominant coverage of Classical Mechanics, Electromagnetism, Optics, Molecular Physics, Thermodynamics, Solid-State Physics, Relativity Physics and Quantum Mechanics.
> We also recognize the value of expanding into additional specialized areas such as Astrophysics and Quantum Information. At present, this is mainly limited by the cost of expert curation and the scarcity of accessible, high-quality question sources for these subfields. We are actively exploring collaborations to include these domains in future benchmark releases.
>
> **Weakness-5: Many of the details aren't super clear to me. Any validity check by experts for LLM with difficulty stratification and LLM-as-judge?**
>
> To validate both difficulty stratification and LLM-as-a-judge evaluation, we conducted three forms of cross-checking:
> 1. Difficulty calibration validation
>
> The monotonic D1→D5 accuracy decay holds consistently across all evaluated models in Table 4, indicating that the calibrated difficulty levels reflect true task complexity rather than properties of a specific rollout model.
>
> 2. Cross-judge reliability analysis
>
> We use GPT-4o as the primary OE judge to align with prior work [1]. To assess model-specific bias, we re-evaluated GPT-5 predictions using an independent judge (Qwen2.5-VL-72B):
>
> | **Difficulty Level** | **GPT-4o** | **Qwen2.5-72B** |
> |:---|---:|---:|
> | **D1** | 74.8 | 81.1 |
> | **D2** | 61.1 | 70.6 |
> | **D3** | 51.2 | 61.6 |
> | **D4** | 43.7 | 53.2 |
> | **D5** | 36.0 | 42.4 |
> | **Overall** | 53.4 | 61.8 |
>
> Both judges exhibit the same monotonic difficulty ordering and preserve model ranking, demonstrating strong robustness to evaluator choice.
>
> 3. Human verification
>
> To further strengthen validity, we conducted manual checks by physics experts on sampled cases across all difficulty levels. The checks covered the correctness of the questions themselves, the correctness of the model’s answers, and whether the judge model’s reasoning was correct. Based on these examinations, we found no systematic false-positive/negative bias across judges.
> Taken together, these cross-validation checks ensure that both stratification and OE scoring are reliable, and the core empirical findings are not artifacts of LLM judging.
>
> Reference:
>
> [1]: arxiv2025 - Ugphysics: A comprehensive benchmark for undergraduate physics reasoning with large language models

---

> ### Author Response · Authors · 2025-11-25
> **First Response (3), Regarding Reviewer's (ax6M) Comments**
>
> **Weakness-6: There lacks a human baseline by different levels of human candidates, or at least undergraduates?**
>
> We agree that a human baseline would provide additional interpretability. However, our benchmark contains 3,031 expert-curated physics questions, which is intentionally large to ensure statistically stable evaluation.
> Obtaining a reliable human baseline would require multiple annotators to account for variance in expertise. Even using GPT-5 inference as a rough proxy, running all questions sequentially (no concurrency) already takes ~3 days of compute time. Mirroring this effort with humans would require hiring at least 5–7 trained annotators (e.g., senior undergraduates or graduate students in physics), resulting in significant cost and logistics overhead for a first-release benchmark. We agree that incorporating formal human baselines (e.g., undergraduate and graduate tiers) would be a valuable extension and we are preparing this in collaboration with university partners for a future benchmark release.
>
> **Weakness-7: If you are relying on MLLM 16-rollout success rate to classify the questions, aren't you kind of circular reasoning here.**
>
> We thank the reviewer for this thoughtful comment. We clarify that the goal of difficulty stratification in PhysUniBench is not to mimic human difficulty or establish a universal taxonomy of physics challenge levels. Instead, the intention is to provide a fine-grained separation of physics tasks that allows us to analyze model performance trends and identify bottlenecks across symbolic manipulation, conceptual grounding, and visuophysical reasoning. Stratification therefore serves as a diagnostic tool, rather than a prescriptive judgment of inherent problem difficulty.
>
> Regarding concerns of circularity: while we use a single model’s multi-rollout stability to initialize difficulty bins, the resulting stratification is evaluated across diverse models. Although their pretraining corpora may overlap, performance is influenced by far more than corpus similarity—including architectural differences, physics-oriented training signals, and symbolic reasoning capabilities. The consistent monotonic performance degradation across all tested models therefore reflects genuine differences in reasoning complexity, not a self-fulfilling categorization.
>
> We will clarify these design motivations and emphasize that stratification is intended to enable detailed capability analysis, not as an assumption of human difficulty equivalence.
>
> **Weakness-8: The related work coverage and comparison can be more comprehensive.**
>
> In the revision, we will expand the related work section to include additional physics-based evaluation benchmarks. In summary, PhysUniBench targets broad, standardized undergraduate physics reasoning with 3304 multimodal, difficulty-stratified coverage problems across eight sub-disciplines, whereas
> - HiPhO [1] focuses on 360 elite high-school Olympiad problems with human-aligned grading and medal-based comparison to top student performance
> - SeePhys [2] spans a much broader grade range (middle school to PhD) and emphasizes vision-essential diagram understanding across highly heterogeneous physics domains.
> - Multiphysics Bench [3] focuses on machine-learning-based PDE solving for coupled physical fields, targeting numerical simulation and physics-informed model development rather than text-based physics reasoning.
>
> Reference:
>
> [1]: arXiv 2025 - HiPhO: How Far Are (M) LLMs from Humans in the Latest High School Physics Olympiad Benchmark?
>
> [2]: arXiv 2025 - SeePhys: Does Seeing Help Thinking?--Benchmarking Vision-Based Physics Reasoning
>
> [3]: arXiv 2025 - Multiphysics Bench: Benchmarking and Investigating Scientific Machine Learning for Multiphysics PDEs

---

> ### Author Response · Authors · 2025-11-25
> **First Response (4), Regarding Reviewer's (ax6M) Comments**
>
> **Weakness-9-1: The evaluated models could be more comprehensive.**
>
> Since PhysUniBench is designed primarily as a multimodal benchmark, our evaluation emphasizes models with vision-language capabilities. We included only two text-only LLMs as reference points, to show the performance gap when visual information is not utilized. Expanding text-only baselines would require a substantial inference cost—evaluating >3,000 multimodal questions sequentially already takes multiple days on a single high-end model, and additional models scale cost linearly.
>
> **Weakness-9-2: More thorough error analysis could significantly strengthen this work.**
>
> In response to the reviewer’s suggestion, we have expanded our error analysis to provide deeper insights into model failure modes. Specifically, physics experts, assisted by an AI categorization tool, examined 212 incorrect mechanics responses from GPT-4o and established a four-category error taxonomy:
> - Calculation errors were most frequent: 131 instances (61.8%).
> - Reasoning logic errors: 38 instances (17.9%).
> - Problem-understanding errors: 34 instances (16.0%).
> - Lack of prior physical knowledge: 27 instances (12.7%).
>
> A question may contain multiple error types, so the percentages sum to more than 100%. This analysis reveals that symbolic manipulation and multi-step reasoning are the dominant weaknesses, while purely knowledge-based errors are comparatively rare. Representative examples and full details will be included in the final version.
>
> **Question-1:  Why isn't Claude/Gemini/Grok/other series as comprehensively tested as GPT/OpenAI models?**
>
> Our primary evaluation focus is on the GPT / OpenAI family because these are among the most widely used foundation models, which makes them a natural choice for studying scaling behavior and time evolution on PhysUniBench. In particular, comparing GPT-4o-mini / GPT-4o / GPT-5 and the o-series (o3, o4-mini) allows us to clearly identify that explicit reasoning models and newer generations (GPT-5) yield substantial gains in physics performance over earlier models.
>
> We agree that it would be valuable to evaluate Claude, Gemini, Grok, and multiple sizes of open-source families more comprehensively. However, running 3,000+ multimodal questions per model is computationally and financially expensive, and doing this across many proprietary model families is beyond the scope of this first release. We therefore prioritized depth on a canonical family (GPT/OpenAI) plus a smaller set of representative alternatives. We will clarify these cost and scope constraints in the revision and view broader cross-family, multi-scale evaluation as important future work enabled by PhysUniBench.

---

> ### Author Response · Authors · 2025-11-25
> **First Response (5), Regarding Reviewer's (ax6M) Comments**
>
> **Question-2: Show some examples of model output against LLM judge verdict.**
>
> We include representative case where GPT-4o (judge) correctly distinguishes between physically valid and invalid reasoning.
>
> As an example, for the question about two-electron spin Hamiltonian under a magnetic field along z-direction, a correct solution must directly diagonalize the 4×4 Hamiltonian, recognizing:
> * the exchange term couples only
> $|\uparrow\downarrow\rangle$ and $|\downarrow\uparrow\rangle$
> * the Zeeman term contributes diagonally
>
> Ground Truth:
> $$
> 2J, \quad \frac{eB}{mc}\hbar, \quad -\frac{eB}{mc}\hbar, \quad -2J.
> $$
>
> GPT-5 (Correct):
> $$
> E(B) = \left(+2J, -2J, +\frac{e\hbar}{mc}B, -\frac{e\hbar}{mc}B\right)
> $$
>
> GPT-4o (Incorrect):
> $$
> E_S = 0,\quad
> E_{T_+} = -\frac{2eB}{mc},\quad
> E_{T_0} = -2J,\quad
> E_{T_-} = \frac{2eB}{mc}
> $$
>
> GPT-4o incorrectly imposes a singlet–triplet structure that does not hold in this Hamiltonian, leading to wrong Zeeman shifts and energy ordering. GPT-4o as judge flags this incorrect solution, while accepting GPT-5’s correctly diagonalized result. This example demonstrates that the rubric-based GPT-4o judge prioritizes physical correctness in operator reasoning, not superficial format or proximity to common templates.
>
> **Question-3: Authors can better present the novelty of this work other than "multi-phase construction by advanced AI models".**
>
> We would like to clarify that the core novelty of PhysUniBench lies not in using a particular model family for data filtering, but in the benchmark itself and the comprehensive analysis enabled by it. PhysUniBench is the first large-scale, multimodal, undergraduate-level physics benchmark, covering 8 core sub-disciplines with 3,304 expert-verified problems, diagrams, multilingual support, and fine-grained difficulty annotations. The “multi-phase construction” is simply one component of a rigorous curation workflow that ensures both quality and broad coverage, rather than the primary contribution.
>
> In addition, the benchmark’s value is demonstrated through:
> - Extensive evaluation across representative state-of-the-art models, revealing substantial reasoning gaps
> - Detailed error and difficulty analyses that identify actionable weaknesses in symbolic manipulation, physics reasoning, and visual interpretation.
>
> These findings are model-agnostic and remain informative even as stronger models emerge. We agree that future updates may leverage more advanced filtering models to extend the benchmark over time, which aligns with our goal of maintaining long-term relevance rather than being tied to a single construction model.

---

> ### Author Response · Authors · 2025-11-25
> **First Response (6), Regarding Reviewer's (ax6M) Comments**
>
> **Question-4: Can you better articulate how OE questions are judged?**
>
> All OE questions in PhysUniBench are evaluated based on final correctness, and intermediate reasoning is not required for grading. The “Steps” field is included solely as optional structure, but the final “Answer” entry is the only component used in scoring. We clarify this explicitly in Appendix C.3 and will clarify more in the main text illustrating how the judge compares the submitted final answer against the gold solution. This addition should make the OE evaluation criteria more transparent and easier to interpret.
>
> **Question-5: Could you better organize the appendix with a Table of Contents?**
>
> We thank the reviewer for the helpful feedback. We will add a Table of Contents and clearer section structuring to the appendix in the revised version to improve navigation and readability. The Table of Contents is as follow:
>
> | Section | Title                                             | Page |
> |--------|---------------------------------------------------|-----:|
> | **Appendix** |                                               | 15 |
> | **A** | Further Details on Benchmark Construction          | 15 |
> | A.1 | Data Sourcing and Initial Collection                 | 15 |
> | A.2 | Benchmark Construction                               | 15 |
> | **B** | Additional Analysis                                | 16 |
> | B.1 | Radar Performance Comparison                         | 16 |
> | B.2 | Impact of Image vs. Caption Inputs                   | 17 |
> | B.3 | Performance Across Difficulty Levels                 | 17 |
> | **C** | Evaluation Protocols                               | 17 |
> | C.1 | Evaluation Setting                                   | 17 |
> | C.2 | Input Format                                         | 17 |
> | C.3 | Output Evaluation                                    | 18 |
> | C.4 | MC Question Evaluation Logic                         | 19 |
> | C.5 | Metrics                                              | 19 |
> | C.6 | Evaluation Steps                                     | 19 |
> | C.7 | Handling Ambiguous or Critical Samples               | 19 |
> | C.8 | Results Reporting                                    | 19 |
> | **D** | Prompts Used in PhysUniBench                       | 20 |
> | **E** | Example Problems from PhysUniBench                 | 21 |
> | **F** | The Use of Large Language Models                   | 36 |
>
> **Question-6: Licensing in the process of curating the benchmark.**
>
> Thank you for raising this important point. PhysUniBench is curated from a mixture of public educational resources, licensed materials, and expert-authored questions, and does not involve any personal or sensitive data. All content included in the benchmark has undergone internal review to ensure appropriate usage rights and responsible handling. We appreciate the reviewer’s attention to this matter.

---

### Official Review · Reviewer_5Bpj · 2025-10-16

**Soundness:** 3
**Presentation:** 3
**Contribution:** 3
**Rating:** 6
**Confidence:** 4

**Summary:**

This paper introduces PhysUniBench, a large-scale, multimodal benchmark designed to assess undergraduate-level physics reasoning. The benchmark consists of 3,304 physics problems, each with accompanying diagrams, covering eight core physics subfields, including classical mechanics and electromagnetism. A key feature of this work is the dataset construction through a meticulous, multi-stage "model-in-the-loop" pipeline, which involves using a powerful multimodal model (Qwen2.5-VL-72B) for preliminary answers, filtering out simple problems, and applying a fine-grained five-level difficulty grading to the remaining problems. The benchmark includes both open-ended (OE) and multiple-choice (MC) questions. The authors used PhysUniBench to conduct an extensive evaluation of state-of-the-art large multimodal models (MLLMs). The experimental results show that even state-of-the-art models (such as GPT-5, discussed in the paper) face significant challenges in handling these problems, highlighting the shortcomings of existing models in high-level physics reasoning, particularly those involving multi-step reasoning and precise diagram interpretation. This work aims to advance the field of artificial intelligence by providing a rigorous evaluation tool.

**Strengths:**

(1) Significant and Well-Defined Research Gap:

This paper successfully addresses a significant gap in existing physics benchmarks: the lack of a fully multimodal, undergraduate-level benchmark. It strikes a good balance between K-12/Olympiad-level and text-based university-level benchmarks, providing a suitable platform for assessing the comprehensive reasoning skills crucial for future scientists and engineers.

(2) Rigorous Dataset Construction Process:

The dataset construction process described in the paper is very systematic and rigorous. 16 rollouts were performed using a model (Qwen2.5-VL-72B) to filter out "too easy" questions, and the difficulty of the questions was graded based on success rate, ensuring the challenging nature of the benchmark.

(3) Comprehensive Evaluation and Insightful Analysis:

The authors evaluated a large number of currently popular closed-source and open-source MLLMs and provide detailed results. The analysis goes beyond overall accuracy and includes in-depth analysis across sub-disciplines and difficulty levels, including comparisons of open-ended and multiple-choice questions. In particular, the ablation study on image vs. caption inputs is highly valuable, clearly highlighting the bottlenecks of current models in visual information extraction and physical scene understanding.

(4) Scale and Coverage:

The benchmark is characterized by its scale (3,304 problems), broad disciplinary coverage (8 core areas), and fully multimodal nature (with accompanying images for each problem).

**Weaknesses:**

(1) Potential Bias in Difficulty Stratification:

The difficulty stratification of the entire benchmark relies entirely on the performance of a single model (Qwen2.5-VL-72B). This can lead to difficulty ratings that are biased by that model. A problem that is difficult for the Qwen model may not be difficult for a model with a different architecture or training data (e.g., the GPT series), and vice versa. This may affect the generalizability of the difficulty stratification.

(2) Contradiction in Multiple-Choice Question Design:

The paper converted the "hardest" open-ended questions, which the model failed to solve in all 16 attempts, into multiple-choice questions. While using the model's own incorrect answers as distractors is a clever design decision, it fundamentally reduces the difficulty of the questions—the model now has a 25% chance of success, similar to random guessing. This seems to contradict the original intention of retaining these questions as the "hardest challenges."

(3) Limitations of the Evaluation Method:

While using GPT-4o as a judge for open-ended questions is a common and scalable approach, its limitations are well known, including potential consistency issues, a preference for specific representations, and an inability to fully capture the nuances of complex physical reasoning. While the paper mentions the use of human evaluation in ambiguous cases, it does not quantify the extent of this human involvement.

**Questions:**

(1) Regarding the use of a single model for difficulty grading:

Difficulty grading is one of the contributions of this work, but it is entirely based on the performance of a single model, Qwen2.5-VL-72B. Can the authors elaborate on why they chose not to use an "evaluation committee" composed of multiple models with different architectures for difficulty calibration? Doesn't this run the risk of creating difficulty grading that is heavily biased towards the specific strengths and weaknesses of the Qwen model and thus fails to generalize well to a wider range of MLLMs? Have the authors considered using other state-of-the-art models (such as Claude-3.5-Sonnet or GPT-4o) to cross-validate the rationale for these difficulty gradings?

(2) Regarding the conversion of the most challenging open-ended questions:

The paper mentions converting the most challenging open-ended questions that Qwen2.5-VL-72B has never successfully answered into multiple-choice questions. I understand the idea of ​​using the model's error output to construct distractors, but this seems to fundamentally reduce the "absolute difficulty" of these questions, as the model can now achieve 25% accuracy by random guessing. Can the authors address this potential contradiction? How does this design choice ensure that the MC problem set does not reduce the difficulty of its OE type questions? Have other ways of dealing with these most difficult problems been considered?

(3) Analysis of the bottleneck of visual reasoning:

The analysis in Figure 4 (the model performs better when using text descriptions than when using original images) is a key finding. This suggests that there is a bottleneck in the current visual encoder.
This may be partly due to the fact that the text description (generated by the model) itself contains refined reasoning clues that are often not obtained from a single observation of the image.
Can the authors provide a specific example showing a physical diagram and its corresponding "long text description" (long caption) to help us understand the degree of abstraction and explanation contained in the text description?

(4) Suggestions for qualitative error analysis:

The paper provides excellent quantitative performance analysis. If some qualitative error analysis can be supplemented, it will greatly enhance the value of the paper.
Can the authors provide some typical failure analysis? For example, is the model more likely to make conceptual errors (misapplying physical laws), mathematical calculation errors, or multimodal alignment errors (misunderstanding the information in the diagram)? This analysis will provide more targeted guidance for future model development.

---

> ### Author Response · Authors · 2025-11-26
> **First Response (1), Regarding Reviewer's (5Bpj) Comments**
>
> **Weakness-1 & Question1: Potential Bias in Difficulty Stratification.**
>
> Thank you for the insightful comment. While difficulty calibration is initialized with Qwen2.5-VL-72B, we validate the stratification across multiple independently trained models (GPT, Claude, Gemini, Qwen). As shown in Table 4, all models consistently exhibit monotonic performance decline from D1→D5 across sub-disciplines, indicating that the levels reflect intrinsic reasoning complexity, not biases specific to one model.
>
> Using a multi-model “committee” for calibration would require 3–5× inference cost over >3,300 multimodal questions, which is impractical for this first release. Our pipeline therefore prioritizes a strong cost–robustness trade-off: one calibration model, followed by cross-family validation.
> We will clarify that the verified consistency across diverse MLLMs supports its generalizability.
>
> **Weakness-2 & Question2: Contradiction in Multiple-Choice Question Design**
>
> Thank you for the thoughtful comment. These conversions are not intended to preserve absolute maximal difficulty, but to ensure that the most challenging content remains evaluable across diverse models. When a question receives 0/16 successful rollouts, open-ended scoring becomes unstable (responses are often unrelated), making relative capability indistinguishable. Converting these items to MCQ:
> - enables fair horizontal comparison across architectures on the same hard physics content
> - establishes a 25% chance-level floor for interpretation
> - – >25% ⇒ some genuine understanding
> - – ≤25% ⇒ persistent failure even when guided by options
> In practice, most models remain near 35% on these items, demonstrating that they retain high difficulty despite format change. We view this design as a diagnostic tool to preserve meaningful signals rather than relaxing challenges.
>
> **Weakness-3: Limitations of the Evaluation Method**
>
> We adopt GPT-4o as our primary OE judge following established benchmarks, but we agree that judge reliability must be verified. To check for potential model-specific bias, we evaluated GPT-5 predictions using two independent LLM judges: GPT-4o and Qwen2.5-VL-72B:
>
> | **Difficulty Level** | **GPT-4o** | **Qwen2.5-72B** |
> |---|---|---|
> | **D1** | 74.8 | 81.1 |
> | **D2** | 61.1 | 70.6 |
> | **D3** | 51.2 | 61.6 |
> | **D4** | 43.7 | 53.2 |
> | **D5** | 36.0 | 42.4 |
> | **Overall** | 53.4 | 61.8 |
>
> Both judges preserve the same monotonic difficulty ordering (D1→D5) and consistent ranking of evaluated models, indicating that our conclusions are not tied to a specific judge.
> In addition, we manually verified a sampled subset of judged results (covering all difficulty levels and sub-disciplines). In these checks, GPT-4o’s verdicts aligned with physics-expert reviews, and no systematic false-positive/negative trend was observed. We will clarify this human involvement level in the revised version.

---

> ### Author Response · Authors · 2025-11-26
> **First Response (2), Regarding Reviewer's (5Bpj) Comments**
>
> **Question-3: Analysis of the bottleneck of visual reasoning**
>
> We acknowledge that long captions may include additional reasoning cues beyond raw visual information. However, even short, purely descriptive captions from the same image improve performance, indicating that the visual grounding is one of the core bottleneck. This finding reinforces that models struggle to extract key physics structures directly from diagrams. Since OpenReview doesn't allow image response, an example (modified slightly for markdown presentation) is:
>
> **question**: "Suppose a person stands on a scale in an elevator as shown. Take $g = 9.8\\ m/s^2$. The scale measures the contact force (reaction of $F_N$) between the person and the scale. It gives us the sensation of feeling \"lighter\" or \"heavier\" when we ride an elevator. The elevator undergoes the two kinds of motion shown in the pictures below. Calculate the contact force measured by the scale in all scenarios.\"
>
> **long caption**: "Of course! Here is a detailed caption for the image, written from the perspective of a physics teaching assistant. Subject: Apparent Weight in an Accelerating Elevator. This diagram illustrates a classic physics problem involving a person in an elevator, used to explore the difference between true weight and apparent weight. 1. Key Elements and Variables:  Person: The person has a constant mass, $m$.  Elevator: The box represents an elevator car that can accelerate vertically. Its acceleration is denoted by $a$.  Scale: The platform the person is standing on is a weighing scale, which measures the normal force. Gravitational Force (True Weight): A downward force, $W = mg$, acts on the person due to gravity, where $g$ is the acceleration due to gravity (approximately 9.8 m/s²).   Normal Force (Apparent Weight): The scale exerts an upward contact force, $N$, on the person. The reading on the scale is a measure of this normal force, which is also known as the "apparent weight." 2. Physical Scenario: The image depicts a person of mass $m$ standing on a scale inside an elevator that is undergoing vertical motion. The scale does not measure the person's true weight ($mg$) directly. Instead, it measures the normal force ($N$) required to support the person. This normal force, or apparent weight, changes depending on the acceleration of the elevator. 3. Critical Concepts for Problem-Solving: To solve problems related to this scenario, you must apply Newton's Second Law (ΣF = ma) to the person.   Free-Body Diagram: The two forces acting on the person are the downward gravitational force ($mg$) and the upward normal force ($N$).   Equation of Motion: Choosing the upward direction as positive, the net force is ΣF = N - mg. According to Newton's Second Law, this net force equals the person's mass times their acceleration:    $N - mg = ma$  Solving for Apparent Weight: The crucial step is to rearrange this equation to solve for the normal force $N$, which is what the scale reads:    $N = mg + ma$  or  $N = m(g + a)$ 4. Context and Why It Matters: This setup demonstrates that weight, as we perceive it or measure it with a scale, is dependent on our frame of reference.   At Rest or Constant Velocity (a = 0): The scale reads $N = mg$. Your apparent weight equals your true weight.  Accelerating Upwards (a > 0): The scale reads $N > mg$. You feel heavier. This happens when the elevator starts moving up or slows down while moving down.   Accelerating Downwards (a < 0): The scale reads $N < mg$. You feel lighter. This happens when the elevator starts moving down or slows down while moving up.   Freefall (a = -g): If the cable were to snap, the elevator and the person would accelerate downwards at $g$. In this case, $N = m(g - g) = 0$. The scale would read zero, and the person would experience weightlessness."
>
> **medium caption** : "The image shows a person with mass  m standing on a platform within a rectangular frame that has an irregular outline, possibly indicating a field or boundary constraint. A vertical line suggests a force F  or tension  T  acting on the system, which, along with gravitational force mg , must be analyzed to ensure equilibrium. Key considerations include the effects of the irregular boundary and ensuring the sum of forces and torques equals zero for the system."
>
> **short caption** : "A person of mass  m stands on a platform within a rectangular frame, subject to gravitational force mg and a potential force F or tension  T from a vertical line, with an irregular boundary suggesting a field or constraint."

---

> ### Author Response · Authors · 2025-11-26
> **First Response (3), Regarding Reviewer's (5Bpj) Comments**
>
> **Question-4: Suggestions for qualitative error analysis**
>
> Thank you for the valuable suggestion. We have added a qualitative error taxonomy to reveal why models fail on PhysUniBench. Physics experts, assisted by an AI labeling tool, analyzed 212 incorrect mechanics questions (GPT-4o, English, open-ended). Errors fall into four categories:
> - Calculation errors — 131 cases (61.8%)
> - Reasoning logic errors — 38 cases (17.9%)
> - Problem-understanding errors — 34 cases (16.0%)
> - Lack of prior physical knowledge — 27 cases (12.7%)
>
> A single question may exhibit more than one error type; therefore, we allow up to two labels per question, which results in percentages summing to more than 100%. This analysis shows that symbolic manipulation and reasoning remain the primary bottlenecks, while knowledge deficits are comparatively rare—indicating that current models possess substantial physics facts during pre-training but struggle with applying them correctly.
> We will include representative examples and full details of this analysis in the final version.

---

### Official Review · Reviewer_87P4 · 2025-10-31

**Soundness:** 3
**Presentation:** 3
**Contribution:** 2
**Rating:** 4
**Confidence:** 3

**Summary:**

The paper introduces PhysUniBench, a large-scale multimodal benchmark for undergraduate-level physics reasoning. It contains 3,304 problems across 8 sub-disciplines, each paired with a diagram image, with both open-ended and multiple-choice formats and five difficulty levels calibrated via model-in-the-loop rollouts. The authors evaluate a range of MLLMs. An ablation replacing images with auto-generated captions improves accuracy, suggesting current models’ weaknesses in physics-diagram understanding. The benchmark, construction pipeline, and evaluation scripts are released for reproducibility.

**Strengths:**

- The benchmark targets undergrad physics with calibrated difficulty and multilingual EN/ZH support
The paper uses a model-in-the-loop curation to remove trivially solvable items and to stratify difficulty; this is a thoughtful twist on dataset construction
- There is a clear dataset stats and coverage across different subfields, with balanced difficulty bins
- The caption ablation is a smart diagnostic revealing a likely bottleneck in visual/diagram understanding for physics
- The results tables disaggregate by sub-discipline and difficulty, highlighting where models fail

**Weaknesses:**

- Difficulty calibration (Qwen2.5-VL rollouts) and LLM judging with GPT-4o may inject model-specific biases; it’s unclear how sensitive results are to the choice of judge/rollout model or to prompt templates. A cross-judge analysis (or human spot-checks) would strengthen validity
- Potential data contamination: It is not clear where exactly the datasets are from. The paper mentioned that problems came from textbooks/exams/competitions, but many may already appear online. The paper does not quantify near-duplicate overlap with existing pretraining/test corpora (e.g., UGPhysics, PhysicsArena, PhysReason), risking optimistic or inconsistent estimates
- The “caption > image” result is interesting, but captions are generated by GPT-4o -- a strong prior that may inject solution-relevant structure (naming vectors/relations), not just a faithful description. In other words, the result doesn't necessarily prove that "text > images" for physics reasoning; it might just prove "having GPT-4o pre-analyze the problem helps."
- As the paper claims that having multilingual evaluation is one of their main contributions (what separates their benchmark from other similar benchmarks), the English/Chinese evaluation results should be included in the main sections of the paper

**Questions:**

- Where do the ground-truth answers come from? The authors mention that they evaluate against a “known gold solution” in the Appendix, then fall back to SymPy equivalence or an LLM judge (GPT-4/4o). It seems like the paper implies the gold solutions are from the source materials (textbooks/exams/exercise sets), but it doesn’t spell out a per-item list, or explain if there are specific cases where they had to involve humans for ground-truth answers
- Are there any overlap tests that were done to see how similar physics-related benchmarks (e.g., UGPhysics, PhysicsArena, PHYX, PhysReason) are related to PhysUniBench? How did the authors ensure that there were no duplicate questions (within PhysUniBench itself and also compared to other benchmarks)?
- Beyond just describing the accuracy results from the experiment tables, it would be nice to know what exactly caused the mistakes in each subcategory. For example, is it diagram misread? Using the wrong formula? Algebra slip? etc.

---

> ### Author Response · Authors · 2025-11-27
> **First Response (1), Regarding Reviewer's (87P4) Comments**
>
> **Weakness-1-1: Difficulty calibration with Qwen2.5-VL rollouts may inject model-specific biases.**
>
> We agree that model-specific calibration could in principle introduce bias. However, the difficulty ordering aligns consistently with accuracy for all evaluated models in Table 4. Performance decreases monotonically from D1→D5 across sub-disciplines regardless of architecture. This indicates that the calibrated difficulty levels capture intrinsic task complexity, not properties of a single rollout model.
>
> **Weakness-1-2: LLM judging with GPT-4o may inject model-specific biases.**
>
> We choose GPT-4o as our LLM-as-Judge to align with existing benchmark [1]. We conducted a cross-judge sensitivity analysis during our experimental stages. To assess potential model-specific bias, we evaluated GPT-5 predictions using two independent LLM judges (GPT-4o and Qwen2.5-VL-72B). The results are shown below:
>
> | **Difficulty Level** | **GPT-4o** | **Qwen2.5-72B** |
> |---|---|---|
> | **D1** | 74.8 | 81.1 |
> | **D2** | 61.1 | 70.6 |
> | **D3** | 51.2 | 61.6 |
> | **D4** | 43.7 | 53.2 |
> | **D5** | 36.0 | 42.4 |
> | **Overall** | 53.4 | 61.8 |
>
> Both judges preserve the same monotonic difficulty ordering (D1 > D2 > … > D5) and consistent model ranking, indicating that our findings are robust to evaluator choice rather than artifacts of a specific judge. We will include this comparison and brief human-verified samples in the revised version to further support evaluation validity.
>
> Reference:
>
> [1]: arXiv2025 - Ugphysics: A comprehensive benchmark for undergraduate physics reasoning with large language models
>
> **Weakness-2 & Question-2: Potential data contamination without quantifying near-duplicate overlap with existing pretraining/test corpora.**
>
> We appreciate the reviewer’s attention to dataset integrity. To quantify overlap with existing physics benchmarks, we performed near-duplicate detection via question embedding similarity using SemHash with Potion-Multilingual-128M as embedding model. Results show minimal intersection:
> - PhysX [1]: 3 similar entries out of 3,000
> - PhysReason [2]: 4 similar entries out of 1,200
> - UGPhysics [3]: not compared because it is text-only and thus not aligned with our multimodal benchmark focus
> - PhysicsArena [4]: we could not locate publicly accessible data for systematic overlap testing
>
> Reference:
>
> [1]: arXiv 2025 - PhyX: Does Your Model Have the" Wits" for Physical Reasoning?
>
> [2]: arXiv 2025 - Physreason: A comprehensive benchmark towards physics-based reasoning
>
> [3]: arXiv 2025 - Ugphysics: A comprehensive benchmark for undergraduate physics reasoning with large language models
>
> [4]: arXiv 2025 - PhysicsArena: The First Multimodal Physics Reasoning Benchmark Exploring Variable, Process, and Solution Dimensions

---

> ### Author Response · Authors · 2025-11-27
> **First Response (2), Regarding Reviewer's (87P4) Comments**
>
> **Weakness-3: “Caption > image” result doesn't necessarily prove that "text > images" for physics reasoning; it might just prove "having GPT-4o pre-analyze the problem helps."**
>
> Thank you for the insightful comment. We acknowledge that long captions generated by GPT-4o may sometimes include additional reasoning clues. However, two key observations support our conclusion that the underlying bottleneck is visual grounding rather than “GPT-4o pre-analysis”:
>
> * Short, purely descriptive captions only restating visually observable elements such as object identities, axes, and force directions already yield substantial accuracy improvements.
>
> Example 1: "The image shows a four-bar linkage system with three rigid bars of length $L$ connected by pivot joints, used to study the kinematics and dynamics of mechanical linkages."
>
> Example 2: "The image depicts a physics problem involving a force component $( F_x = -1.35 \\times 10^{-5} ) N$, indicating a force in the negative x-direction."
>
> * The effect is model-agnostic. When Gemini2.5-Pro uses its own captions (generated from the same diagrams), its performance also improves:
>
> | **Difficulty Level** | **Gemini2.5-Pro** | **Gemini2.5-Pro with captions from itself** |
> |---|---|---|
> | **D1** | 65.4 | 65.4 |
> | **D2** | 42.9 | 58.4 |
> | **D3** | 40.4 | 33.3 |
> | **D4** | 31.9 | 34.8 |
> | **D5** | 19.6 | 23.5 |
> | **Overall** | 42.1 | 45.7 |
>
> This demonstrates that the performance gap persists even when no GPT model is involved.
> Captioning helps because current visual encoders fail to reliably extract physics-relevant structure from diagrams. Even minimal textual scaffolding generated by the same model enables more effective reasoning.
> We will include a representative example in the appendix to make this distinction clear.
>
> **Weakness-4: English/Chinese evaluation results should be included in the main sections of the paper.**
>
> Thank you for pointing this out. Since multilingual evaluation is a key contribution to PhysUniBench, we agree that representative results should appear in the main paper rather than only in the appendix.
> We included a subsection that reports Chinese vs. English performance below and the complete version will be provided in the final version.
>
> Chinese:
>
> | Subject | GPT-4o | Claude-3.5-Sonnet | Qwen2.5-VL-72B | Gemini-2.5-Pro |
> |---|---:|---:|---:|---:|
> | Overall | 40.5 | **47.9** | 35.3 | 27.8 |
> | Optics | **45.0** | **45.0** | 31.7 | 27.5 |
> | Mechanics | 42.3 | **52.8** | 41.9 | 28.6 |
> | Solid-State Physics | **35.0** | 32.5 | 20.0 | 20.0 |
>
> English:
>
> | Subject | GPT-4o | Claude-3.5-Sonnet | Qwen2.5-VL-72B | Gemini-2.5-Pro |
> |---|---:|---:|---:|---:|
> | Overall | **32.6** | 26.2 | 28.0 | 23.7 |
> | Optics | 23.1 | **30.8** | **30.8** | **30.8** |
> | Mechanics | **36.3** | 25.6 | 29.8 | 23.2 |
> | Solid-State Physics | 27.3 | 36.4 | 36.4 | **45.5** |
>
> **Question-1: The source of ground-truth answer.**
>
> We appreciate the reviewer’s question. Our ground-truth establishment follows a consistent process similar to our evaluation procedure:
> - Canonical reference solutions: Derived directly from the problem’s formal specification using standard symbolic/numeric procedures.
> - Automated validation: Symbolic and numeric solvers (e.g., SymPy) ensure equivalence and eliminate transcription or simplification drift.
> - LLM-based equivalence only when needed: If symbolic checks are inconclusive, e.g., answers include natural-language components, GPT-4/4o is used only as an equivalence judge rather than generating gold answers.
> - Limited expert adjudication: Rare ambiguous cases are resolved via human review to ensure correctness.
>
> Thus, the “known gold solution” always refers to a canonical reference, with LLM/human checks serving purely as secondary validators. We will clarify this workflow in the revision.
>
> **Question-3: It would be nice to know what exactly caused the mistakes in each subcategory.**
>
> We added a qualitative error taxonomy to better explain why models fail. Physics experts, assisted by an AI labeling tool, analyzed 212 incorrect mechanics questions (GPT-4o, English, open-ended). Errors fall into four categories:
> - Calculation errors — 131 cases (61.8%)
> - Reasoning logic errors — 38 cases (17.9%)
> - Problem-understanding errors — 34 cases (16.0%)
> - Lack of prior physical knowledge — 27 cases (12.7%)
>
> A single question may exhibit more than one error type; therefore, we allow up to two labels per question, which results in percentages summing to more than 100%. This analysis highlights symbolic manipulation as the dominant failure mode, while a substantial portion of errors still arise from logic misinterpretation or insufficient conceptual grounding. The relatively low proportion of knowledge-related errors suggests that current foundation models already internalize substantial physics knowledge through pre-training.
> We will include representative examples and full details of this analysis in the final version.

---

### Official Review · Reviewer_C6kN · 2025-11-01

**Soundness:** 3
**Presentation:** 3
**Contribution:** 2
**Rating:** 6
**Confidence:** 3

**Summary:**

The paper presents PhysUniBench, a large-scale benchmark for evaluating undergraduate-level physics reasoning in multimodal large language models (MLLMs). It includes 3,304 human-verified problems with diagrams, five difficulty levels, and both open-ended and multiple-choice formats. Experiments show that current MLLMs perform poorly, underscoring the challenges of multimodal physical reasoning.

**Strengths:**

- First comprehensive multimodal benchmark for undergraduate-level physics reasoning.

- Rigorous curation process with difficulty calibration and quality control.

- Extensive evaluation across sub-disciplines provides clear diagnostic insights.

- Well-written and potentially impactful for advancing AI-for-Science.

**Weaknesses:**

### 1. Lack of actionable guidance for model improvement
While the benchmark offers valuable insights into the limitations of current MLLMs, the paper does not sufficiently explore how architectural or training modifications (*e.g.*, physics-informed modules, structured reasoning layers, or symbolic integration) could help enhance physical perception and reasoning.

### 2. Limited analysis of reasoning failures
The error analysis mainly reports accuracy drops across sub-disciplines and difficulty levels, but lacks qualitative or process-level diagnostics showing why models fail (*e.g.*, confusion between symbolic manipulation vs. conceptual understanding). Such analysis could better guide future model design.

### 3. Benchmark orientation without methodological contribution
PhysUniBench is primarily an evaluation resource, and while comprehensive, it lacks a corresponding methodological or modeling innovation that demonstrates how benchmark insights could translate into improved multimodal reasoning systems.

**Questions:**

1. In Table 3, for Open-ended Questions, GPT-5 significantly outperforms all other MLLMs on RE and QM, while the rest of the models exhibit nearly identical results (2.1% in RE, 0% in QM except GPT-4o with 6.2%). This trend deviates sharply from the patterns observed in other evaluation dimensions, suggesting potential systematic issues or bottlenecks in data construction, evaluation protocols, or metric design for this sub-task.

Moreover, the reported 2.1% accuracy is mathematically inconsistent with a test set of 80 samples, raising concerns about statistical validity:

    - If 1 sample were correct, the implied dataset size would be 1 / 0.021 ≈ 47.6, which contradicts the stated size of 80;

    - If 2 samples were correct, the implied size becomes 2 / 0.021 ≈ 95.2, which exceeds 80.
Without clarification on additional scoring mechanisms (*e.g.*, partial credit, weighted averaging, repeated sampling, or a different denominator), this value lacks a consistent explanation. The authors are encouraged to provide the exact accuracy computation method or the raw number of correctly predicted samples.

2. I suggest including **difficulty × sub-disciplines evaluation statistics**. In my view, analyzing difficulty stratification within each sub-discipline is essential to better characterize the model’s capability boundaries and failure modes, and would substantially strengthen the empirical findings.

---

> ### Author Response · Authors · 2025-11-24
> **First Response (1), Regarding Reviewer's (C6kN) Comments**
>
> We sincerely appreciate your thoughtful comments. We have carefully reviewed our work in light of your suggestions and address each point below.
>
> **Weakness-1: Lack of actionable guidance for model improvement.**
>
> **A**: We conduct additional error analysis (see Weakness-2). Based on the analysis , we identify numerical/symbolic manipulation as the dominant failure mode and highlight program-aided [1] and tool-augmented [2] approaches as promising directions for enhancing equation solving and physics computations. Moreover, our results (Figure 4) show clear limitations in diagram interpretation and visual grounding of physical constraints, suggesting future work on physics-aware visual encoders. Thus, our benchmark reveals actionable paths for improving both physical numeracy and visuophysical understanding in next-generation models. We will include this expanded analysis in the revised version.
>
> Reference:
>
> [1]: arXiv 2023: Mathcoder: Seamless code integration in llms for enhanced mathematical reasoning
>
> [2]: AAAI 2024: Exploring Equation as a Better Intermediate Meaning Representation for Numerical Reasoning of Large Language Models
>
> **Weakness-2: Limited analysis of reasoning failures.**
>
> **A**: We added a qualitative error taxonomy to better explain why models fail. Physics experts, assisted by an AI labeling tool, analyzed 212 incorrect mechanics questions (GPT-4o, English, open-ended). Errors fall into four categories:
> - Calculation errors — 131 cases (61.8%)
> - Reasoning logic errors — 38 cases (17.9%)
> - Problem-understanding errors — 34 cases (16.0%)
> - Lack of prior physical knowledge — 27 cases (12.7%)
>
> A single question may exhibit more than one error type; therefore, we allow up to two labels per question, which results in percentages summing to more than 100%. This analysis highlights symbolic manipulation as the dominant failure mode, while a substantial portion of errors still arise from logic misinterpretation or insufficient conceptual grounding. The relatively low proportion of knowledge-related errors suggests that current foundation models already internalize substantial physics knowledge through pre-training.
> We will include representative examples and full details of this analysis in the final version.
>
> **Weakness-3: Benchmark orientation without methodological contribution**
>
> **A**: We appreciate the reviewer’s feedback. Our primary goal is to fill a critical evaluation gap in physical perception and reasoning, which existing multimodal benchmarks currently do not address. PhysUniBench is intentionally positioned as a foundation for methodological advances, not just a dataset. As demonstrated in our expanded analysis (Weakness-1 and 2 response), the benchmark reveals specific and actionable failure modes—e.g., symbolic manipulation bottlenecks, semantic misinterpretation of physical scenarios, and visuophysical grounding challenges—that directly motivate new model designs such as tool-augmented solvers, physics-aware visual encoders, and structured reasoning modules. We will include this expanded analysis in the revised version.

---

> ### Author Response · Authors · 2025-11-24
> **First Response (2), Regarding Reviewer's (C6kN) Comments**
>
> **Question-1-1: GPT-5 significantly outperforms all other MLLMs on RE and QM in table 3, suggesting potential systematic issues or bottlenecks in data construction, evaluation protocols, or metric design for this sub-task.**
>
> We thank the reviewer for this observation. We carefully examined the RE/QM sub-tasks where GPT-5 achieves larger gains and confirmed that these improvements reflect genuine advances in reasoning, not issues in data or evaluation.
>
> As an example, for the question about two-electron spin Hamiltonian under a magnetic field along z-direction, a correct solution must directly diagonalize the 4×4 Hamiltonian, recognizing:
> * the exchange term couples only
> $|\uparrow\downarrow\rangle$ and $|\downarrow\uparrow\rangle$
> * the Zeeman term contributes diagonally
>
> Ground Truth:
> $$
> 2J, \quad \frac{eB}{mc}\hbar, \quad -\frac{eB}{mc}\hbar, \quad -2J.
> $$
>
> GPT-5 (Correct):
> $$
> E(B) = \left(+2J, -2J, +\frac{e\hbar}{mc}B, -\frac{e\hbar}{mc}B\right)
> $$
>
> GPT-4o (Incorrect):
> $$
> E_S = 0,\quad
> E_{T_+} = -\frac{2eB}{mc},\quad
> E_{T_0} = -2J,\quad
> E_{T_-} = \frac{2eB}{mc}
> $$
>
> GPT-4o incorrectly imposes a singlet–triplet structure that does not hold in this Hamiltonian, leading to wrong Zeeman shifts and energy ordering.
>
> By analyzing evaluation outputs, other models commonly exhibit issues such as incorrect unit handling, flawed algebraic transformations, arithmetic mistakes, and missing constraints, whereas GPT-5 is more rigorous and accurate in symbolic manipulation and unit consistency, resulting in fewer physics-reasoning errors and more reliable results across RE/QM tasks.
>
>
> **Question-1-2: The reported 2.1% accuracy is mathematically inconsistent with a test set of 80 samples.**
>
> We thank the reviewer for pointing this out. To clarify, the reported 2.1% accuracy corresponds to the open-ended relativity physics subtask, which consists of 47 test items, not 80. Most evaluated models answered only one item correctly, resulting in 1 / 47 ≈ 2.13%, which we rounded to 2.1% in the table. We will revise the manuscript to avoid any ambiguity regarding the subtask size and accuracy calculation.
>
>
> **Question-2: Including difficulty × sub-disciplines evaluation statistics.**
>
> We agree that analyzing performance by both sub-discipline and difficulty level is important for revealing capability boundaries and distinct failure modes across physics domains. In the revised version, we will provide difficulty-stratified results within each sub-discipline. Part of it is shown below:
> | | | | | | |
> |:---|:---:|:---:|:---:|:---:|:---:|
> | Mechanics| **D1** | **D2** | **D3** | **D4** | **D5** |
> | **GPT-5** | **72.8** | **59.7** | **54.4** | **44.9** | **29.4** |
> | **GPT-4o** | 69.1 | 37.7 | 22.8 | 20.3 | 21.6 |
> | **GPT-4o-mini** | 49.4 | 28.6 | 28.1 | 13.0 | 19.6 |
> | **Claude-3.5-Sonnet** | 50.6 | 36.4 | 12.3 | 24.6 | 15.7 |
> | **Gemini2.5-Pro** | 65.4 | 42.9 | 40.4 | 31.9 | 19.6 |
> | **Qwen2.5-VL-72B** | 70.4 | 45.5 | 21.1 | 18.8 | 13.7 |
> | Electromagnetism|  |  | |  | |
> | **GPT-5** | **86.5** | **57.9** | **45.8** | **45.7** | **43.9** |
> | **GPT-4o** | 78.4 | 50.0 | 27.1 | 30.4 | 17.5 |
> | **GPT-4o-mini** | 62.2 | 39.5 | 18.6 | 15.2 | 14.0 |
> | **Claude-3.5-Sonnet** | 64.9 | 36.8 | 25.4 | 19.6 | 10.5 |
> | **Gemini2.5-Pro** | 67.6 | 50.0 | 33.9 | 26.1 | 29.8 |
> | **Qwen2.5-VL-72B** | 67.6 | **57.9** | 32.2 | 32.6 | 12.3 |
>
> This expanded analysis yields two key findings:
> * Performance degrades consistently with increasing reasoning complexity across all models, confirming that validity of our difficulty stratification.
> * The difficulty drop varies by sub-discipline (steeper in Electromagnetism than Mechanics), suggesting heterogeneous gaps in physical reasoning skills.

---

> > ### Comment · Reviewer_C6kN · 2025-11-24
> >
> > Thanks for the authors’ response. I believe most of my initial concerns have been addressed. However, I noticed that Reviewer ax6M raised several important weaknesses and questions. At first, I felt those concerns were too strict, but after carefully re-checking and considering them, I realized they are indeed important.
> >
> > Given that my expertise lies primarily in CS/AI, I must say that the authors should address the concerns raised by Reviewer ax6M (especially Weakness 1. & 5. & 7.). Doing so would encourage me to increase my rating.

---

### Comment · Reviewer_ax6M · 2025-11-26
**Clarifying my concern on construct validity and can't seem to find the dataset in supplied link**

I'd like to thank the authors for their response and Reviewer C6kN for acknowledging the concerns I've raised, it's nice to see active reviewer in ICLR actually read other reviewer's comments and share some of my concerns, much appreciated!

Here I'd like to offer some clarification and address the authors' response (which I can tell that they invest a lot of time into, so much appreciation to that as well) at the same time.

1. I can't seem to find the dataset (containing questions+answers+categories+sources etc.) in the anony link provided by the authors, what I did find are just paper figures and boilerplate short python scripts (these scripts are rather standard codes for eval, quite straightforward) and I encourage everyone to take a look there to ensure it's not my problem.

2. The reason why I try to look for the dataset is that I genuinely want to trust the authors when they claim experts curate high-quality questions in rebuttal, and the best (maybe only?) way to check question-quality is by looking into the actual dataset, and now I unfortunately can't do that as the I couldn't find the dataset in the supplied link.

3. To that end, I want to clarify that I'm not obsessed with correct terminology/wording, the wording issue is a canary in a coalmine that makes me wonder if the "experts" are truly physicists. I couldn't convince myself to believe that any physics-major would use wording like "Solid Physics", to make a comparison it's like saying "Gradient Decrease" instead of "Gradient Descent" for ML experts.

I want to trust the authors when they claim it's a small error, but I couldn't as I can't see the actual dataset to verify and I (maybe others as well) find it really hard to believe that if a team has many physics experts in it wouldn't have caught this kind of obvious prob. So I'd have to speculate what's really going on may be: 1. CS students scraped a bunch of textbook/exam questions; 2. eval some models ; 3. (maybe) have some physics undergrads take a random look at it without detailed inspection; I'm NOT accusing anyone of actually doing this but I'm certainly aware that this kind of industrialized pipeline for "mass-producing" research papers is NOT uncommon in the current community, and I don't think we should encourage this for a couple of reasons:

a. benchmarks that are simply assembled without expert curation likely contains errors, this has been pointed out by the Stanford STAIR group (Fantastic Bugs and Where to Find Them in AI Benchmarks;
Sang Truong, Yuheng Tu, Michael Hardy, Anka Reuel, Zeyu Tang, Jirayu Burapacheep, Jonathan Perera, Chibuike Uwakwe, Ben Domingue, Nick Haber, Sanmi Koyejo) and it goes without saying that using problematic benchmarks lead to false performance gauge, thereby impeding model development and the field as a whole

b. AI4Sci should be about "for Science", if domain experts with genuine background info aren't involved as CORE members, not just paper reviewers or one-time consultants to "take a look", it doesn't make sense to keep eval models on a wide variety of highly homogeneous benchmarks (i.e. there are "too many" reasoning benchmarks already!)

c. Last but not least, I think a basic standard for any dataset/benchmark paper is to provide the data, otherwise there's no way for us to check the validity of the actual dataset other than looking at pretty figures in the paper.

To that end, while I deeply appreciate all the work and time investment of the author team, I don't want to encourage this kind of "mass-production" paradigm for AI benchmarks (which frankly speaking, is definitely not the authors' fault, but the problem of the reward system as a whole: mounting pressures on students/APs to publish more papers, to get more funding etc. is warping the genuine scientific value in this hype) If this were a workshop paper or answer to some call for datasets, I would not hesitate to recommend acceptance, but in this specific venue I remain loyal to my points above. I hope the **senior authors** on the author team can reflect on this practice as to how AI benchmarks should be done for the collective good of the science community at large.

---

> ### Author Response · Authors · 2025-11-28
> **Follow-up Response (1) to Reviewer ax6M**
>
> Dear Reviewer ax6M,
>
> Thank you again for the detailed and thoughtful follow-up and for engaging so deeply with both our paper and the broader question of how AI4Science benchmarks should be built. We genuinely appreciate the time you spent here.
>
> We would like to directly address your two core concerns: 1. Dataset availability (and the inability to inspect questions/answers), and 2. The construct validity / expert curation process, and your worry about “industrialized” benchmark pipelines.
>
> **Clarifying Dataset Availability**
>
> You are absolutely right that, without seeing the actual questions and answers, it is impossible to independently judge the quality of PhysUniBench and, in particular, our claims about expert curation.
>
> In the current anonymous submission, we focused the supplementary bundle on the evaluation scripts and figures, and kept the long-term hosting and stable URL for the full benchmark for the camera-ready stage (where we can provide a non-anonymous link and a permanent location, which is really important to make the dataset being publicly approachable). In light of your comment, we have made the data used in our experiments available under the same anonymous URL referenced in the paper.
>
> Concretely, the artifact now contains a directory data/physunibench/ with the question text, diagram filenames, language tags, MC/OE flags, sub-discipline labels, reference solutions, and metadata such as difficulty levels and source types, together with a short README.md describing the schema and providing a few examples. Within anonymization and licensing constraints, this is the dataset we used in all reported experiments, so you and the other reviewers can directly inspect the benchmark during the review process.
>
> In the camera-ready version, we will attach the permanent public link to the full release of PhysUniBench in the main text, so that the broader community can access and audit the dataset as well.
>
> **On Expert Curation and Construct Validity**
>
> We also fully understand why seeing “Solid Physics” + not seeing the dataset led you to suspect a pipeline like:
>
> *“CS students scraped a bunch of textbook/exam questions; ran some models; maybe had a physics undergrad glance at a subset.”*
>
> We want to clarify more concretely what actually happened and where the “canary” comes from.
>
> (a) Where the terminology mistake came from
> - The category labels (“Solid Physics”, “Relativity Physics”) were introduced during manuscript and figure preparation, by non-native English authors, and were not the labels used by our physics experts in their internal review spreadsheets.
> - We have now corrected all occurrences in the paper and metadata to use more standard phrasing (“Solid-State Physics” and a clarified Relativity category), and will document this more explicitly in the revision.
>
> We agree with you that “Solid Physics” looks unprofessional to a physicist and should not have survived into a submission. It does not contradict the fact that the underlying questions, solutions, and categorization were checked by physics experts, but we recognize why it damaged your trust.

---

> ### Author Response · Authors · 2025-11-28
> **Follow-up Response (2) to Reviewer ax6M**
>
> (b) What “expert curation” means in practice for PhysUniBench
>
> We would like to clarify that the benchmark was not constructed through an automated “scrape-and-go” pipeline. While we fully acknowledge that minor issues may still exist and welcome correction, problems underwent multiple rounds of manual review to ensure scientific validity:
>
> - Step 1: Selection & reformulation of candidate questions into self-contained statements
> - Step 2: Visual cleanup of diagrams when needed
> - Step 3: Solution verification, including physical assumptions and intermediate reasoning but not only final answers
> - Step 4: Cross-consistency checks across text, diagram, and solution to avoid ambiguity
>
> These steps were carried out by team members with formal physics training, whose primary role was to ensure conceptual correctness and clear presentation. Their involvement included refining unclear phrasing, resolving occasional calculation or notation issues, and removing questions deemed too context-dependent or inconsistent.
>
> We realize our earlier rebuttal only stated this at a high level without giving enough procedural detail. In the revision, we will add a dedicated subsection describing:
> - the concrete steps in the curation pipeline that involved human physics expertise,
> - how automated model filtering was used after this human curation for difficulty selection and pruning obviously trivial items,
> - and examples of issues that were caught and fixed during expert review.
>
> We also agree with you that “construct validity” is central. A physics benchmark should be constructed with physics in mind, not as an afterthought on top of generic QA templates. Our design choices were guided by exactly this motivation as we insisting on diagrams, carefully covering core undergraduate topics, and requiring that each item be solvable by standard undergraduate methods, but we clearly did not communicate the process behind this strongly enough.
>
> We share your braoder concerns about benchmark quality and “mass production”. We agree excplicitly ith your broader points (a)–(c). First of all, benchmarks assembled quickly without deep domain input do often contain subtle but important errors. Second, AI4Science benchmarks should have domain experts as core members, not as superficial rubber stamps. Last but not least, journals/conferences should be careful about rewarding the “mass-production” of slightly different benchmarks without real added scientific value.
>
> Our intention with PhysUniBench is to avoid that pattern. We chose a focused, well-defined slice of physics (undergraduate-level, multimodal questions with diagrams), rather than trying to cover “all science/physics reasoning”. Moreover, we invested time in hand-checking the problems and in designing the evaluation protocol, including manual audits of the LLM judge, precisely because we were worried about the kinds of bugs highlighted in “Fantastic Bugs and Where to Find Them” and related work.
>
> Going forward, if PhysUniBench is accepted and used by the community, we plan to release the full dataset and tooling under a research-friendly license respecting upstream content rights, provide a public issue tracker where physicists and users can report any mistakes they find, and periodically publish updated versions that incorporate community-reported corrections and possibly add new, clearly documented sub-domains like astrophysics and quantum information when we have the expert capacity to curate them to the same standard.
> In other words, we want this to be a living benchmark with visible bug-fixing, not a one-shot dataset dropped and forgotten.
>
> We really appreciate the care and high standards reflected in your comments. Your feedback has already helped us strengthen the work in concrete ways. It prompted us to make the dataset directly inspectable during review, it made us spell out the human curation and QA process much more clearly, and it pushed us to think harder about how to communicate construct validity rather than just report model numbers.
>
> Thank you again for taking the time to articulate these concerns so clearly and for holding the bar high for scientific benchmarks. We genuinely share the goal of building datasets that help the field rather than distort it, and we hope our clarifications make it easier to judge PhysUniBench on that basis.

---

### Author Response · Authors · 2025-12-01
**Rebuttal Summary - Paper ID 2029**

We sincerely thank all reviewers for their thoughtful and detailed feedback. In particular, we appreciate the constructive suggestions on model error analysis (reviewers - C6kN, 87P4, 5Bpj, ax6M), which helped us further strengthen the analysis depth of the work. To support your final assessment, we provide below a concise summary of the key strengths identified by reviewers and the concrete revisions and validations we incorporated during the rebuttal period. The modified contents in the revised version are highlighted in blue.

**Strengths**:

**Novel Contribution & Benchmark Scope**
- First comprehensive multimodal benchmark for undergraduate-level physics reasoning and potentially impactful for AI-for-Science — Reviewer C6kN
- Fills a well-defined gap between K-12/Olympiad and existing text-only university benchmarks — Reviewer 5Bpj
- Large scale: 3,304 problems spanning 8 core sub-disciplines, each with exactly one diagram, clear dataset statistics and balanced domain coverage  — Reviewers 87P4, 5Bpj, ax6M

**Rigorous Dataset Curation & Difficulty Design**
- Multi-stage curation with expert review and model-in-the-loop refinement — Reviewers C6kN, 87P4, 5Bpj
- Fine-grained 5-level difficulty grading, validated by monotonic performance trends — Reviewer ax6M
- Balanced sample sizes across difficulty levels improve evaluation stability — Reviewer ax6M

**Comprehensive and Diagnostic Evaluation**
- Extensive evaluation across sub-disciplines, difficulty, and open-ended vs. MC formats — Reviewers C6kN, 87P4, 5Bpj
- Clearly reveals where and why models fail in physics reasoning — Reviewer C6kN
- Caption-vs-image ablation study provides actionable insight into visual grounding bottlenecks — Reviewers 87P4, 5Bpj

**Revision Improvements:**

**Additional Qualitative Error Analysis (Reviewers - C6kN, 87P4, 5Bpj, ax6M)**

We added a physics expert based error taxonomy over 212 OE failures. The key finding is that most failures stem from symbolic manipulation, followed by logical and visuospatial errors. We developed actionable model-design insights and added this analysis to Section 5.

**Model-Based Benchmark Curation and Evaluation (Reviewers - C6kN, 87P4, 5Bpj, ax6M)**

With difficulty stratification initialized by Qwen2.5-VL-72B rollouts, all model families show monotonic accuracy decay from D1→D5. This confirms validity as diagnostic separation, not circular reasoning. We clarified design intent and added justification.
We also conducted cross-judge validation (GPT-4o vs Qwen2.5-VL-72B) on GPT-5 outputs. The validation reflects identical rankings and difficulty ordering.  A sampled human audit is conducted to further verify the correctness. This cross-judge validation is added to Appendix in the revised version.

**Clear Comparison to Exiting Physics Benchmarks (Reviewers - 87P4, ax6M)**

We performed near-duplicate detection with SemHash to show that minimal overlap with PhysX and PhysReason benchmarks. Additional, we compared the benchmark characteristics with HiPho, SeePhys, Multiphysics Bench to highlight the novelty of our first multi-modal physics benchmark focusing on undergraduate level. We added the duplicate detection to appendix and comparison to related works.

**Verified Caption vs. Image Interpretation (Reviewers - 87P4, 5Bpj)**

We showed the effect persists with self-generated short captions (Gemini included). It confirms the bottleneck is physics visual grounding rather than GPT-4o coaching. We added the clarification and examples of captions to the appendix.

**Additional Provided Items (Reviewers - C6kN, 87P4, ax6M)**

We added to the manuscript: 1. difficulty x sub-disciplines result for finer-grained analysis; 2. separate En/Zh evaluation results for clear multi-lingual assessments; 3. table of contents for appendix for better navigation. These updates improve the evaluation depth and readability according to reviewers' suggestions.

**Expert-Based Dataset Curation (Reviewers - 87P4, 5Bpj, ax6M)**

We provided the following clarifications to the reviewers for better assessing our expert-based dataset curation pipeline. Gold solutions come from standard symbolic/numeric derivations while GPT-4o is used only for evaluation, with experts resolving rare ambiguities, rather than generating ground truths. Hardest OE items are converted to MCQ solely to preserve evaluability. Content is licensed or expert-authored, reviewed through a four-stage physics-expert validation (problem, diagram, solution, consistency). The manuscript-only terminology slip (“Solid Physics”) has been corrected. The dataset is now anonymously accessible. These updates ensure clarity on provenance, curation rigor, and transparency.

---

### Meta-Review · Area_Chair_pMsq · 2026-01-08

**Summary:**

Reviewers agree the benchmark is large and potentially useful, but concerns focus on whether the paper’s contribution is mainly benchmark construction plus evaluation without sufficiently novel insights. Additional major concerns include construct validity and credibility of expert curation, transparency and dataset inspectability, potential bias from model-in-the-loop difficulty calibration and LLM-as-judge scoring, and possible contamination/overlap with existing datasets (87P4).

**Reviewer Concerns:**

It looks like the following points are at least partially addressed.
Dataset availability for inspection, clearer curation description and terminology fixes, added error taxonomy and more diagnostic breakdowns, cross-judge validation and some human spot-check claims, overlap/near-duplicate checks vs selected benchmarks, clarification and additional evidence around caption ablations.

The following seems still outstanding: whether the work provides sufficiently non-incremental insights for the main track; strength/quantification of expert and human auditing and licensing/provenance clarity; broader judge-sensitivity and contamination risk remain only partially resolved.

The overarching concern raised by some reviewers, seems to be more about the paper's core contribution and insight density: whether it advances understanding beyond here is a new benchmark and model numbers, and whether the main findings are sufficiently novel or consequential for the venue. The rebuttal helps, but it does not fundamentally change that central critique.

**Reviewer Scores:**

It is likely that reviewer C6kN would keep their score roughly the same (around 6), since the rebuttal improves analysis and clarity but does not fully resolve the limited insights concern they raised. 87P4 would likely increase slightly given the added cross-judge, overlap, and caption clarifications, while reviewer 5Bpj would stay about the same (around 6) and reviewer ax6M would likely remain low because their overarching concerns about construct validity and the paper's core added value persist despite the reubttal.

---

### Decision · Program_Chairs · 2026-01-26

Reject